# Inference of transcription factor binding from cell-free DNA enables tumor subtype prediction and early detection

Peter Ulz [1], Samantha Perakis [1], Qing Zhou [1], Tina Moser [1], Jelena Belic [1], Isaac Lazzeri[1], Albert Wölfler[2], Armin Zebisch [2], Armin Gerger[3], Gunda Pristauz[4], Edgar Petru[4], Brandon White[5], Charles E.S. Roberts[5], John St. John[5], Michael G. Schimek[6], Jochen B. Geigl [1], Thomas Bauernhofer[3], Heinz Sill[2], Christoph Bock [7,8,9], Ellen Heitzer [1,10,11]* & Michael R. Speicher [1,10]*

Deregulation of transcription factors (TFs) is an important driver of tumorigenesis, but non-invasive assays for assessing transcription factor activity are lacking. Here we develop and validate a minimally invasive method for assessing TF activity based on cell-free DNA sequencing and nucleosome footprint analysis. We analyze whole genome sequencing data for >1,000 cell-free DNA samples from cancer patients and healthy controls using a bioinformatics pipeline developed by us that infers accessibility of TF binding sites from cell-free DNA fragmentation patterns. We observe patient-specific as well as tumor-specific patterns, including accurate prediction of tumor subtypes in prostate cancer, with important clinical implications for the management of patients. Furthermore, we show that cell-free DNA TF profiling is capable of detection of early-stage colorectal carcinomas. Our approach for mapping tumor-specific transcription factor binding in vivo based on blood samples makes a key part of the noncoding genome amenable to clinical analysis.

[1] Institute of Human Genetics, Diagnostic and Research Center for Molecular BioMedicine, Medical University of Graz, Graz, Austria. [2] Department of Internal Medicine, Division of Hematology, Medical University of Graz, Graz, Austria. [3] Department of Internal Medicine, Division of Oncology, Medical University of Graz, Graz, Austria. [4] Department of Obstetrics and Gynecology, Medical University of Graz, Graz, Austria. [5] Freenome, South San Francisco, CA, USA. [6] Institute of Medical Informatics, Statistics and Documentation, Medical University of Graz, Graz, Austria. [7] CeMM Research Center for Molecular Medicine of the Austrian Academy of Sciences, Vienna, Austria. [8] Department of Laboratory Medicine, Medical University of Vienna, Vienna, Austria. [9] Max Planck Institute for Informatics, Saarland Informatics Campus, Saarbrücken, Germany. [10] BioTechMed-Graz, Graz, Austria. [11] Christian Doppler Laboratory for Liquid Biopsies for Early Detection of Cancer, Graz, Austria. *email: ellen.heitzer@medunigraz.at; michael.speicher@medunigraz.at

Transcription factors (TFs) modulate the expression of their target genes and often play a key role in development and differentiation[1]. TF binding is often correlated with nucleosome occupancy[1–4]. It has recently been shown that nucleosome positioning can be inferred from cell-free DNA (cfDNA) in plasma[5], suggesting that it may be possible to infer not only gene expression[6] but also TF binding in tumor samples from circulating tumor DNA. Accurate inference of TF binding from cfDNA would have substantial diagnostic potential in cancer and potentially other diseases, but to date TFs cannot be evaluated noninvasively.

We thus develop an approach capable of providing insights into single TFs directly from nucleosome footprints to objectively assess and compare TF-binding site (TFBS) accessibility in different plasma samples. To validate our method, we produce deep whole-genome sequencing (WGS) data for 24 plasma samples from healthy donors, i.e., subjects without known current disease, and for 15 plasma samples of patients with metastasized prostate, colon or breast cancer, where cfDNA also comprises circulating tumor DNA (ctDNA)[7–10]. Moreover, we generate shallow WGS data for 229 plasma samples from patients with the aforementioned tumor entities (>18.5 billion mapped plasma sequence reads in total) and we furthermore use 769 plasma samples from patients with colon cancer ($n = 592$) and healthy controls ($n = 177$) (~238 billion mapped plasma sequence reads in addition).

Our approach allows identification of lineage-specific TFs and profiling of individual TFs from cfDNA. We demonstrate two relevant clinical applications: first, our TF-based cfDNA assays are capable of distinguishing between prostate adenocarcinoma and small-cell neuroendocrine prostate cancer, a distinction that has important therapeutic implications. Second, the large colon cancer cohort enabled us to accurately establish resolution limits and to explore the use of TF-based plasma analyses for detection of early cancer stages.

## Results

**Bioinformatic cfDNA fragment analysis to infer TF activity.**
We employed stringent criteria and restricted our analyses to high-confidence 504 TFs where 1000 TFBSs per TF were supported by the majority of tissue samples in the Gene Transcription Regulation Database (GTRD; Version 18.01; http://gtrd.biouml.org)[11] (Methods; Supplementary Data 1).

We then extracted tissue and cancer type-specific peak sets from ATAC-seq-based chromatin accessibility data sets[12,13] for hematopoietic lineages, prostate adenocarcinoma (PRAD), breast cancer (BRCA), and colon adenocarcinoma (COAD) and calculated individual TF accessibilities (see the Methods section). As expected, we observed different tissue-dependent TF accessibility patterns (Fig. 1a). For example, accessibility of GRHL2, a pioneer TF for epithelial cells[14] was increased in PRAD, BRCA, and COAD as compared with hematologic lineages, whereas the hematopoietic lineage TF LYL1 (lymphoblastic leukemia 1)[15] showed a reverse pattern with an increased accessibility in hematopoietic lineages (Fig. 1a). In contrast, the well-established prostate lineage TFs AR and NKX3-1[16–19] showed preferential accessibility in PRAD (Fig. 1a). We then compared hematopoietic and epithelial lineages with each other and established a ranking order of TF accessibilities for each epithelial tissue (Supplementary Data 2). For example, the aforementioned TFs AR or NKX3-1 scored top ranks in the PRAD list, whereas hematopoietic lineage TFs such as purine-rich box 1 (PU.1)[20], LYL1, and the lymphocyte lineage-restricted transcription factor SPIB[21] occupied low ranks in all epithelial samples (Supplementary Data 2). As a result, we had a comprehensive list of high-confidence TFs and their respective accessibilities in three different epithelial lineages.

We started our plasma DNA analyses by establishing nucleosome occupancy maps at TFBSs using high-coverage cfDNA samples from 24 healthy controls (males and females, 12 each), where the vast majority (>90%) of cfDNA is derived from apoptosis of white blood cells with minimal contribution from other tissues[22,23] and 11 plasma samples derived from seven patients with three common tumor entities (Supplementary Table 1). These included four cases with prostate cancer (P40, P147, P148, and P190), one colorectal adenocarcinoma (C2), and two breast cancers (B7 and B13) with ctDNA fractions ranging from 18 to 78% (Supplementary Fig. 1; Supplementary Data 3).

We found evidence that nucleosome plasma footprints are informative regarding TFBSs. For example, in healthy individuals, binding sites of the hematopoietic TF LYL1 were surrounded by arrays of strongly positioned nucleosomes yielding a high-amplitude oscillating coverage pattern (Fig. 1b). However, the ctDNA in the plasma from patients with cancer altered the balance between DNA from hematopoietic versus epithelial cells visible as decreased amplitudes for LYL1 (Fig. 1b). Conversely, the amplitude of the epithelial TF GRHL2 was increased in samples from cancer patients compared with those derived from healthy controls (Fig. 1b). In addition to the correlation with the ATAC-seq data, we confirmed the lineage specificity of these TFs with data of publicly available DNase hypersensitivity assays (Fig. 1b). For further confirmation, we also conducted comparisons with ENCODE data, where mononucleosome-bound DNA fragments were generated by micrococcal nuclease (MNase) digestion (Supplementary Fig. 2a, b). Using high-molecular-weight DNA as a negative control, we did not observe a nucleosome-associated uneven coverage pattern at TFBSs (Supplementary Fig. 2c).

We next tested whether TF accessibilities as established from ATAC-seq data can be inferred from the cfDNA nucleosome occupancy patterns. However, currently no means of assessing TF accessibility and changes thereof in cfDNA exist. To implement such an approach, we first investigated TF-specific nucleosome coverage profiles, which led us to conduct calculations separately for TFBSs within and outside of TSSs (Supplementary Fig. 3). These analyses suggested that average TFBS patterns comprise two signals: a TSS-proximal (within 2 kb of TSS, resulting in a low-frequency pattern) and a TSS-distal (>2 kb away from TSS peak, generating a high-frequency pattern), corresponding to the more evenly spaced peak signal (Fig. 1c). To suppress effects on the coverage not contributed by preferential nucleosome positioning and to remove local biases from the nucleosome data, we used Savitzky–Golay filters for detrending (Methods) (Fig. 1c). Subsequently, we recorded the data range (maximum minus the minimum of the data values, corresponds to the amplitude) of the high-frequency signals as a measure for the accessibility of each TFBS. We refer to these rank values as accessibility score.

To benchmark the performance of Savitzky–Golay filtering, we used cfRNA data[24] and observed significantly reduced accessibility for unexpressed TFs (i.e., <0.1 FPKM [fragments per kilobase exon per million reads]; $n = 91$) as compared with the accessibility of expressed (i.e., >10 FPKM; $n = 137$) TFs ($p = 1.254 \times 10^{-11}$; two-tailed Mann–Whitney U test) (Fig. 1d). These differences were also significant when we compared the accessibility scores to mean DNase coverage ($p < 2.2 \times 10^{-16}$; two-tailed Mann–Whitney U test) (Fig. 1e).

**The accessibility score enables accurate inference of TFBSs.** We then used overall z-scores, i.e., the rank differences between a tumor sample and the healthy controls, and defined detection

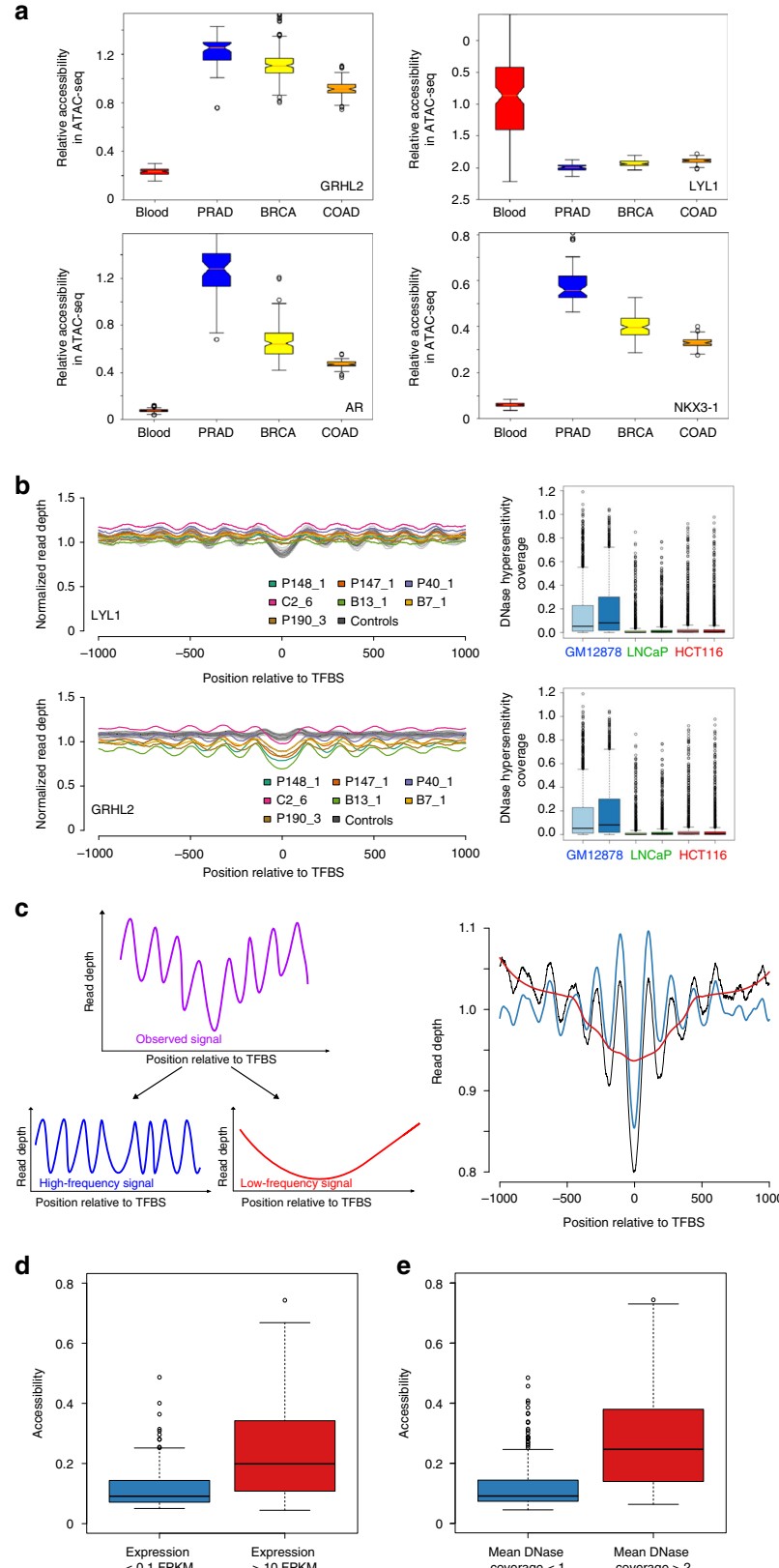

thresholds for TFBS accessibility differences from the mean of rank in normal samples by >5 standard deviations (i.e., a z-score of 5 and −5, respectively). Histogram analysis and a QQ-plot showed no relevant deviation from a normal distribution (Supplementary Fig. 4), suggesting that z-scores are in general applicable for our analyses.

Whereas samples from healthy donors showed no TFs exceeding the ±5 z-score threshold (Fig. 2a), we observed very different patterns in samples derived from patients with cancer. For example, in prostate sample P40_1, TFs with accessibilities above the +5 z-score threshold included, in addition to GRHL2, FOXA1, which cooperates with nuclear hormone receptors in

**Fig. 1** Bioinformatic fragment analysis of cfDNA enables accurate inference of TF activity. **a** Relative accessibility for TFs GRHL2, LYL1, AR, and NKX3-1 for hematopoietic, PRAD, BRCA, and COAD lineages generated from publicly available ATAC-seq data[12,13]. **b** Nucleosome position profiles from plasma DNA for the hematopoietic lineage TF LYL1 and the epithelial TF GRHL2 (left panels; profiles of healthy controls are shown in gray, and patient-derived profiles are displayed in the indicated colors). In plasma samples from patients with cancer, the amplitudes of LYL1 and GRHL2 are decreased and increased, respectively, reflecting the different contributions of DNA from hematopoietic and epithelial cells to the blood. The lineage specificity was further confirmed by DNA hypersensitivity assays (right panels). **c** To measure TF accessibility, the observed raw coverage signal (purple in left and black in right panel) was split by Savitzky–Golay filtering into a high-frequency signal (blue) and a low-frequency signal (red) using different window sizes. The right panel illustrates an overlay of these three signals. The high-frequency signal is used as a measure for accessibility. **d** TFs with increased expression ($n =$ 137; >10 FPKM [fragments per kilobase exon per million reads]) in blood have a significantly higher accessibility as compared with TFs with no or low signs of expression ($n = 91$; <0.1 FPKM). **e** Transcription factors with a mean DNase hypersensitivity coverage of >2 in GM12878 DNase data from the ENCODE project ($n = 213$) have higher accessibilities than factors that have a mean coverage of <1 ($n = 244$). Boxplots are defined as follows: Centerline represents the median, the box represents the interquartile range (IQR), and the whiskers denote the first quartile –1.5 IQR and the third quartile +1.5 IQR, respectively. Notches in (**a**) represent the confidence interval around the median

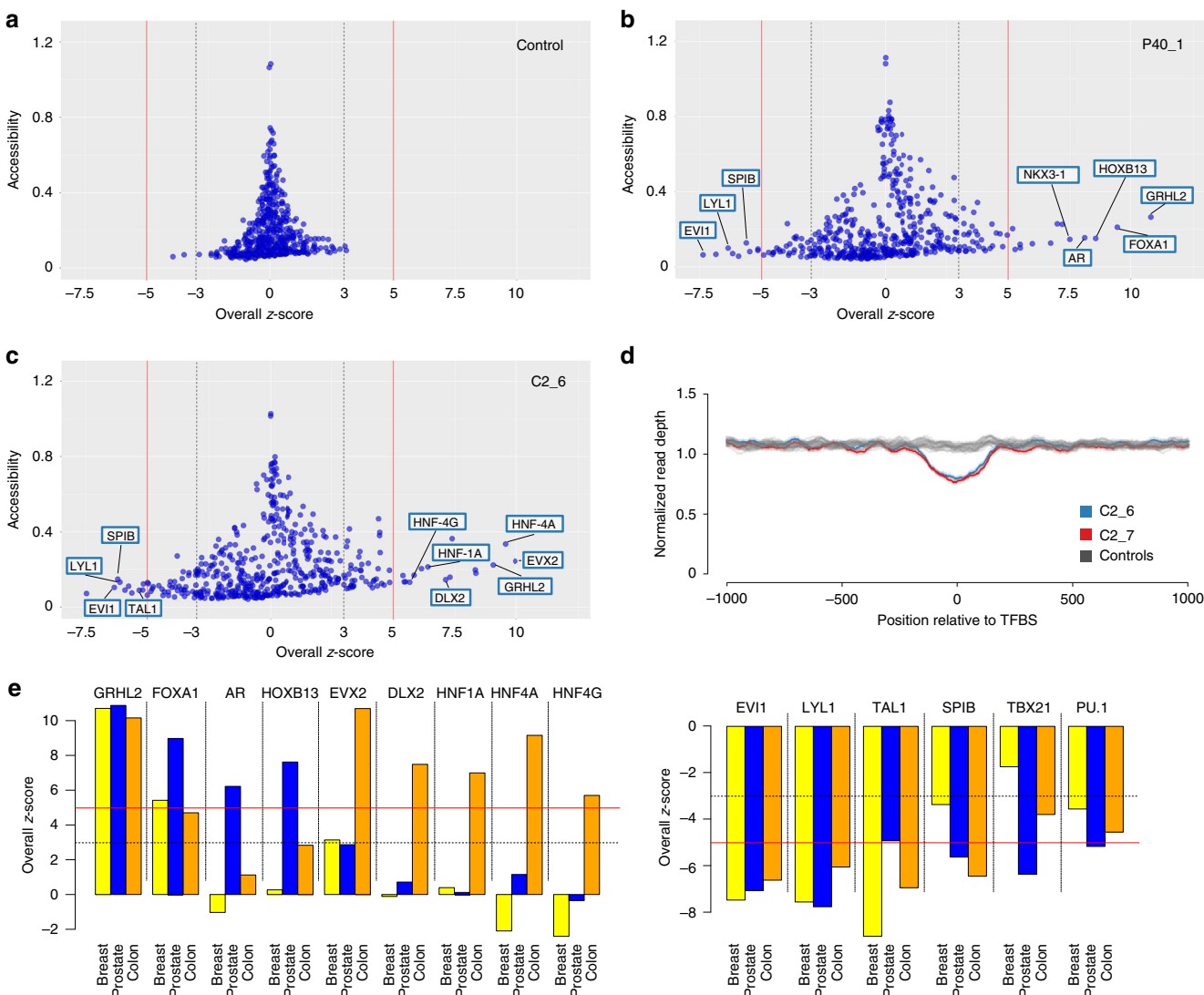

**Fig. 2** Identification of TFs with altered accessibility in plasma samples from patients with cancer. **a** TFBS analysis of a plasma sample from a healthy donor (NPH001). Each point represents a TF, the y-axis displays the accessibility values, and the x-axis illustrates the overall z-score, as a measure of deviation in accessibility from normal control samples. In the samples from healthy donors (compared with every remaining healthy donor), only a few TFs exceeded a z-score of ±3 (dotted gray lines) and none the ±5 z-score (red lines) threshold. **b** An overall z-score plot as in (**a**), but with a plasma sample derived from a patient with prostate cancer (P40). **c** Overall z-score plot as in (**a**) for plasma sample C2_6. **d** Nucleosome position profiles from plasma DNA of healthy controls (gray profiles) and two plasma samples derived from a patient C2 with colon cancer (blue and red) for TF EVX2. **e** Bar charts of overall z-score plots for merged breast, prostate, and colon pools. The left panel displays TFs with increased accessibility in at least one tumor entity; the right panel summarizes the accessibilities of hematopoietic-related TFs. Bars represent single data points

endocrine-driven tumors of the prostate and breast[25], as well as the prostate lineage-specific TFs HOXB13, AR, and NKX3-1[16–19] (Fig. 2b). In contrast, hematopoietic TFs, such as LYL1, SPIB, and EVI1 (transcriptional regulator ecotropic viral integration site 1[26] (Fig. 2b) had low accessibilities. These results were in excellent agreement to our TF ranking based on the ATAC-seq data (Supplementary Data 2; Supplementary Data 4).

In breast cancer samples B7 and B13, we detected in concordance with the ATAC-seq data (Supplementary Data 2; Supplementary Data 4) increased accessibility for GRHL2, FOXA1, and ZNF121 (Supplementary Fig. 5), a zinc finger protein, which was recently implicated in regulation of cell proliferation and breast cancer development[27].

In the samples from colon cancer patient C2, we made a particularly interesting observation. Surprisingly, the ATAC-seq data[13] had ranked EVX2, a TF that has not been intensively studied and has not yet been discussed within the context of cancer[28], as most accessible in COAD (Supplementary Data 2). Indeed, EVX2 was ranked with the highest accessibility in our analysis (Fig. 2c), and the nucleosome position map showed an enormously increased accessibility of EVX2 (Fig. 2d). In agreement with the ATAC-seq data (Supplementary Data 2), we also observed increased accessibility for the TFs HNF4A, GRHL2, DLX2, HNF4G, and HNF1A (Fig. 2d).

Furthermore and as predicted by our evaluation of the ATAC-seq data (Supplementary Data 2), the accessibilities for hematopoietic-related TFs, such as LYL1, TAL1 (SCL/TAL1 (stem cell leukemia/T-cell acute lymphoblastic leukemia (T-ALL) 1))[29], EVI1, TBX21 (T-bet[30]), and PU.1 were reduced in all tumor samples (Fig. 2b, c).

As a further confirmation for the robustness and reproducibility of lineage-specific TFs in cfDNA, we reasoned that if we analyze pools of multiple cfDNA samples generated by shallow-coverage (<0.2×)[31], that those TFs with increased accessibility in the majority of samples should have an increased accessibility score, whereas others will be averaged out. To this end, we pooled cfDNA samples separately for prostate ($n = 69$), for colon ($n = 100$), and for breast ($n = 60$) cancer cases (Supplementary Data 3). When we repeated the analyses, the epithelial TF GRHL2 and hematopoietic TFs reiterated their increased and decreased, respectively, accessibility patterns in the three epithelial lineages (Fig. 2e; Supplementary Data 4). In the colon cfDNA pool, TFs EVX2, DLX2, HNF1A, HNF4A, and HNF4G and TFs AR and HOXB13 in the prostate cancer cfDNA pool had increased accessibilities (Fig. 3e), whereas FOXA1 exceeded the >5 z-score threshold in both the prostate and breast pool. This confirmed that TF accessibility estimation derived from ATAC-seq data (Supplementary Data 2) can be reliably inferred from plasma DNA nucleosome mapping.

**TF analysis predicts prostate cancer tumor subtypes**. To address the question whether TF accessibility remains stable over time, we analyzed two samples each from 4 patients (P40, P147, P148, C2; details on selection of these cases in the Methods section) and found no changes in three of the four plasma sample pairs (P40, P147, and C2) (Supplementary Fig. 6).

However, case P148 showed substantial TFBS accessibility changes in the two analyzed plasma samples (P148_1 and P148_3) (Fig. 3a). Within 12 months, the time interval between collection of these samples, the prostate adenocarcinoma transdifferentiated to a treatment-emergent small-cell neuroendocrine prostate cancer (t-SCNC)[32], which was accompanied by a decrease of PSA (prostate-specific antigen) and an increase of NSE (neuron-specific enolase)[33]. The t-SCNC is no longer an androgen-dependent stage of prostate cancer[34] and, accordingly,

accessibility of AR binding sites was substantially reduced in sample P148_3 (Fig. 3a, left panel). Furthermore, the change in the cell-type identity became apparent as reduced accessibilities to the binding sites of HOXB13, NKX3-1, and GRHL2 (Fig. 3a, left panel). Neuroendocrine reprogramming is facilitated by down-regulation of repressor element-1 (RE-1) silencing transcription factor (REST)[35] and we indeed observed decreased accessibility of REST (Fig. 3a, b). To employ very stringent criteria for assessment of changes in TFBS accessibility, we also calculated z-scores in a pairwise comparison between P148_1 and P148_3. Whereas accessibility of hematopoietic TFs remained unchanged, GRHL2 and NKX3-1 exceeded the −3 z-score and AR, HOXB13, and REST exceeded the −5 z-score (Fig. 3a, right panel). These observations suggested that cancer disease stages with high TFBS plasticity affecting pathways exist.

This case prompted us to assess—as proof of concept—to what extent TFs are suitable for molecular prostate cancer subtyping. The transdifferentiation of an adenocarcinoma to a t-SCNC is a frequent (~20%) mechanism in the development of treatment resistance and has clinically significant implications because it requires a change in therapy[32]. Furthermore, the involvement of TFs in this transdifferentiation process has been extensively studied[34,36,37] (Fig. 3c).

Therefore, we added plasma samples from four further clinically proven t-SCNCs cases (P170_2, P179_4, P198_5, and P240_1) (Supplementary Table 1). For these cases, we additionally tested whether our approach is also applicable to cfDNA sequenced with lower coverage by downsampling plasma samples P148_1 (819,607,690 reads) and P148_3 (768,763,081 reads) to ~50 million reads. The reduction of reads resulted in an increase in noise levels, but this was dependent on the number of TFBSs and therefore negligible for the 504 TFs used in our study, as they had more than 1000 TFBSs (Supplementary Fig. 7). We then repeated the analyses for the aforementioned four samples, each sequenced with ~50 million reads. When we compared these t-SCNC cases with the prostate adenocarcinoma cases, we observed different accessibilities for the TFs AR, HOXB13, NKX3-1, and GRHL2 (Fig. 3d). Interestingly, we noted decreased accessibility of REST in two of these four cases (P170_2 and P198_5; Fig. 3d), which is consistent with reports that REST downregulation is usually observed in 50% of neuroendocrine prostate cancers[34]. These proof-of-concept analyses suggest that inference of transcription factor binding from cfDNA may enable tumor subtype prediction.

**Resolution limits and TF-based early cancer detection**. To quantify the sensitivity of our method, we applied it to 592 plasma samples from individuals with colon cancer, where 82.1% of patients had stage I ($n = 197$) or stage II ($n = 280$) disease (Supplementary Table 2) and compared those to 177 plasma samples from subjects with no current cancer diagnosis (mean read count per sample ~299.2 million, standard deviation 100.4 million reads; mean coverage: ~14.96× with SD 5.02×). Importantly, stage I and II colon cancer represent stages where the tumor is still localized, and most patients can be cured by surgery alone[38].

We employed the ichorCNA algorithm[39] to estimate the tumor content of each sample and found that the vast majority of samples had a tumor content well below ichorCNA's detection limit of 3% (Supplementary Fig. 8). Next, we compared accessibilities in subsamples with different tumor fractions for TFs selected based on our previous ATAC-seq and nucleosome plasma mapping, i.e., GRHL2, EVX2, DLX2, HNF4A, LYL1, and PU.1 (Fig. 3c, e). We observed already statistically significant different accessibilities between healthy controls and COAD

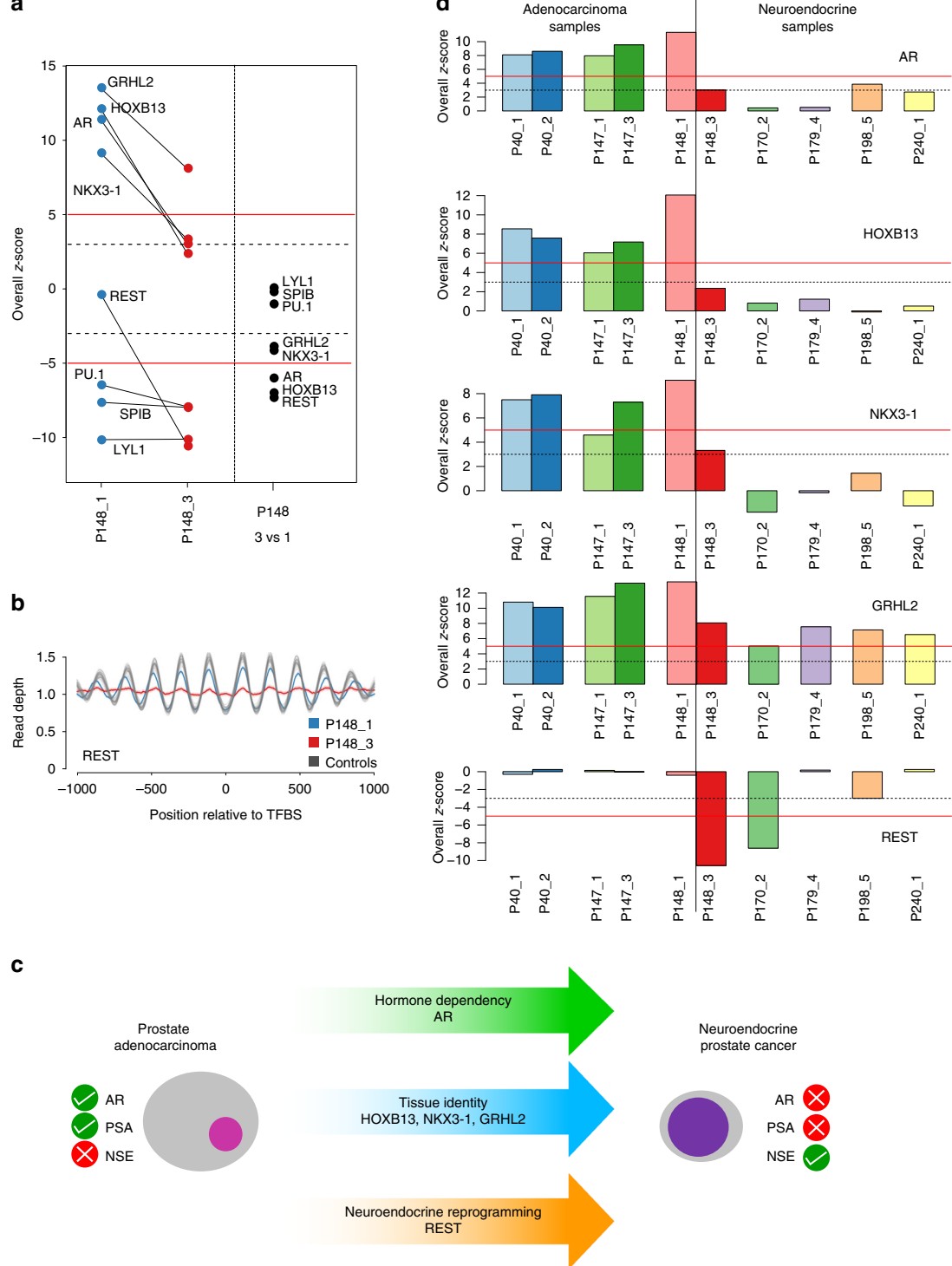

**Fig. 3** Prostate lineage-specific TFs, their plasticity and suitability for tumor classification. **a** Left panel: z-score plots of P148_1 and P148_3 showing drastic accessibility reductions for TFs GRHL2, HOXB13, AR, NKX3-1, and REST whereas the accessibility of hematopoietic TFs PU.1, SPIB, and LYL1 remained unchanged. The right panel displays the z-scores for the pairwise comparison between P148_3 and P148_1. In this pairwise comparison, the hematopoietic TFs had a z-score around 0, whereas GRHL2 and NKX3-1 exceeded −3 and AR, HOXB13, and REST exceeded −5 thresholds. **b** Nucleosome position profiles for TF REST illustrating compared with normal samples a similar accessibility of REST in P148_1 (blue), but a drastic reduction of accessibility in plasma sample P148_3 (red). **c** Prostate adenocarcinomas are AR-dependent and accordingly have frequently increased PSA (prostate-specific antigen) levels and normal NSE (neuron-specific enolase) values. In contrast, t-SCNC are no longer dependent on AR and usually have low PSA and increased NSE levels. Several TFs involved in the transdifferentiation process from an adenocarcinoma to a t-SCNC have been identified and are indicated in the arrows. **d** Bar charts of overall z-score plots for TFs AR, HOXB13, NKX3-1, GRHL2, and REST for prostate adenocarcinomas (left) and prostate neuroendocrine cancer (right). Bars represent single data points

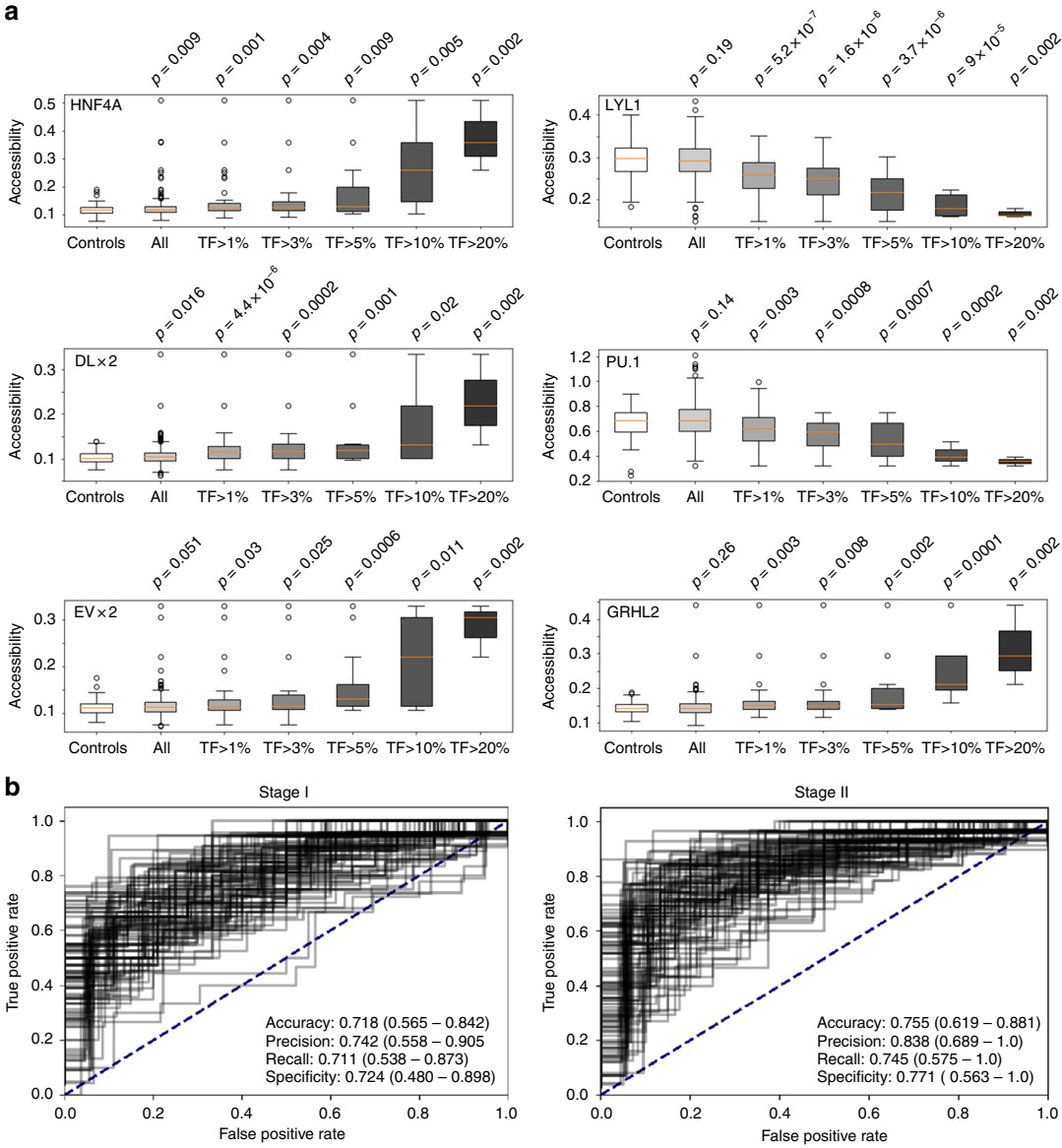

**Fig. 4** TF-based plasma resolution limits and early cancer detection. **a** Comparisons of accessibilities for selected TFs in subsamples of the COAD cohort based on their tumor fraction. Boxplots are defined as follows: centerline represents the median, the box represents the interquartile range (IQR), and the whiskers denote the first quartile −1.5 IQR and the third quartile +1.5 IQR, respectively. **b** Logistic regression with all 504 TFs for samples from the colon cancer cohort with stage I (left panel) and stage II (right panel), respectively. All presented results are cross-validated test-set values

samples for TFs HNF4A and DLX2 (Fig. 4a) and the other TFs, GRHL2, EVX2, LYL1, and PU.1 showed statistically significant differences at a 1% tumor level (Fig. 4a).

We then used logistic regression with all 504 TFs to classify healthy controls from cancer samples. To this end, in 100 permutations, we split the samples into two separate sets and trained the model on one set (90% of samples), while performance metrics were calculated on the withheld samples (10%). We applied this approach to all available samples and to those samples derived from patients with stage I and stage II disease combined (Supplementary Fig. 9; Supplementary Table 3). To test the capability of this approach for detection of early-stage cancer more precisely, we then repeated the logistic regression separately for plasma samples each derived from patients with stage I ($n = 197$) and stage II ($n = 280$). Using our cross-validated test-set values, we found that this approach is capable of identifying ctDNA in patients with stage I colon cancer with a 74% precision (95% CI: 0.53–0.90), 71% sensitivity (recall) (95%

CI: 0.54–0.87), and 72% specificity (0.48–0.90, 95% CI)) (Fig. 4b, left panel), and in patients with stage II colon cancer with a 84% precision (95% CI: 0.69–1.0), 74% sensitivity (recall) (95% CI: 0.58–0.88), and 77% specificity (0.56–1.0, 95% CI)) (Fig. 4b, right panel).

## Discussion

We developed an approach and bioinformatics software pipeline to establish a metric, i.e., the accessibility score, for inferring TF binding from cfDNA in the blood, with relevance for clinical diagnostics and noninvasive tumor classification. While most studies have adopted a gene-centric focus when evaluating somatically acquired alterations[10], we evaluated an important part of the noncoding genome, focusing on TFBSs. As many TFs bind preferentially within open chromatin and have to therefore interact with nucleosomes[1,13], we utilized the largely mono-nucleosomal cfDNA, which allows the mapping of nucleosome positions[5,6]. Our data correlated strongly with ATAC-seq

data[12,13] and DNase I hypersensitivity data for cell lines GM12878, LNCaP, or HCT116, suggesting the reliability of our approach.

Compared with a previous publication by Snyder et al.[5], our study has several distinct differences. First, Snyder et al. profiled a small number of ubiquitous TFs ($n = 8$)[5], whereas we utilized recently generated ATAC-seq data[12,13] and profiled numerous individual TFs, which enabled us to establish lineage-specific TFs for clinical applications. Second, we introduce a metric, the accessibility score, which, by enabling the objective comparison of TF binding events in various plasma samples, paves the way for entirely new diagnostic procedures. In fact, the accessibility score represents a measure of the strength of nucleosome phasing at the binding sites of a TF and reflects the strength of TF binding. Third, we were able to use cfDNA to show TFBS plasticity during a disease course, such as reprogramming to a different cell lineage in prostate cancer. Such a dynamic view of TF activity (vis-à-vis the static view obtained from tissue[13]) is a unique feature of cfDNA analyses. Fourth, we demonstrate that our cfDNA TFBS bioinformatics pipeline allows subclassification of tumor entities, and hence may address an important diagnostic dilemma in the managing of patients with prostate cancer[32]. Fifth, whereas Snyder et al. required 1.5 billion reads per plasma sample[5], which is prohibitive for routine clinical use from a cost-perspective, we were able to conduct in-depth TF analysis with ~50 million reads, making our approach more amenable to clinical applications. Finally, we used a large cohort of patients with colon cancer to establish the resolution limits and provide evidence that our approach is suitable for detection of early-stage cancer. As such, our approach may enable the detection of cancer at stages when the disease is most treatable/curable and thereby help to reduce cancer mortality rates[40].

At the same time, cancer tests should have a high specificity to avoid potentially unsettling healthy individuals with a false positive test result, who would then be subjected to unnecessary follow-up procedures. At present, early cancer detection using liquid biopsies is in its infancy[41] and comparisons with other studies are limited by differences in methodology and study design. The recent CancerSEEK study assessed levels of circulating proteins and mutations in plasma and included 388 patients with colorectal cancer, with a lower percentage of stage I and stage II patients than our study (CancerSEEK: 69%; our study: 82%)[42]. Whereas we achieved a better sensitivity (CancerSEEK: 65%; our cohort: 71 and 74%), the CancerSEEK study reported a high specificity for the entire study cohort of 99%[42]. Another study, which employed size profiling of ctDNA molecules and machine learning, described a lesser specificity than ours of 65% in samples from low-ctDNA cancers (pancreatic, renal, and glioma)[43]. In future applications, it will be interesting to test whether multiparameter strategies, combining different, orthogonal approaches, will result in an improved sensitivity and specificity of plasma-based early cancer detection[10].

Another limitation is that our TF nucleosome interaction maps are inevitably heterogeneous, comprising signals of all cell types that give rise to cfDNA. Furthermore, using all 504 TFs in our logistic regression model does not make our strategy specific for colon cancer. Further work will be required to identify distinct TFs subsets that are specific for different tumor types.

Nevertheless, tumor studies lack dynamic models, and, in particular, dynamic profiling of clinical samples, for exploring transitions and interplays between pathways. Because of the potential of TFs regulating gene transcription throughout the genome and their often exquisite lineage specificity, their detailed noninvasive analyses based on cfDNA offer a unique opportunity to improve clinical diagnostics. Our data also provide the foundation for further dissection of the noncoding genome through means of transcription regulation profiling.

## Methods

**Patients**. The study was approved by the Ethics Committee of the Medical University of Graz (approval numbers 21-227 ex 09/10 [breast cancer], 21-228 ex 09/10 [prostate cancer], 21-229 ex 09/10 [colorectal cancer], and 29-272 ex 16/17 [high-resolution analysis of plasma DNA]), conducted according to the Declaration of Helsinki and written informed consent was obtained from all patients and healthy probands, respectively.

Retrospective human plasma samples from patients with colon cancer (Freenome cohort) were acquired from five biobanks for patients diagnosed with COAD and healthy controls. All plasma samples used in this analysis were de-identified prior to receipt, with no key available to re-identify.

**Blood sampling and library preparation**. Peripheral blood was collected from patients with metastatic prostate, breast, and colon cancer at the Department of Oncology and from anonymous healthy donors without known chronic or malignant disease at the Department of Hematology at the Medical University of Graz. CfDNA was isolated from plasma using the QIAamp Circulating Nucleic Acids kit (QIAGEN, Hilden, Germany) in accordance with the manufacturer's protocol. Regarding library preparation for WGS, we generated shotgun libraries using the TruSeq DNA Sample preparation Kit (Illumina, San Diego, CA, USA) according to the manufacturer's protocol with the following modifications: (i) we used 5–10 ng of input DNA; (ii) due to the length of ctDNA we omitted, the fragmentation step; (iii) for selective amplification of the library fragments, we used 20–25 PCR cycles. Libraries were quality checked on a Bioanalyzer (Agilent) and quantified using qPCR[31].

For the Freenome cohort, cfDNA was extracted from 250 μl plasma using the MagMAX cfDNA Isolation Kit (Applied Biosystems) and converted into libraries using the NEBNext Ultra II DNA Library Prep Kit (New England Biolabs).

Tumor fraction was estimated in each sample using ichorCNA[39] from read counts in 50-kilobase (Kb) bins across the entire genome. In addition 50 kb-bin counts were GC-normalized and normalized by a panel of normal samples using tangent normalization.

**Sequencing**. Control and high-coverage tumor samples were sequenced on the Illumina NovaSeq S4 flowcell at 2 × 150 bp by the Biomedical Sequencing Facility at CeMM, Vienna, Austria. For the control samples, an average of 435,135,450 (range: 352,904,231–556,303,420) paired-end reads were obtained. For the tumor samples (P40_1, P40_2, P147_1, P147_3, P148_1, P148_3, C2_6, C2_7), an average of 688,482,253 reads (range: 541,216,395–870,285,698) were sequenced. Additional samples were sequenced using the Illumina NextSeq platform (B7_1, B13_1, P190_3, P170_2, P179_4, P198_5, P240_1; average sequencing yield: 195,425,394 reads; range: 115,802,787–379,733,061).

Low-coverage tumor samples which were used to create single-entity pools were sequenced on either the Illumina Next-Seq or MiSeq platform. This resulted in 382,306,130 reads from 69 prostate cancer samples; 254,490,128 reads from 60 breast cancer samples and 604,080,473 reads from 100 colon cancer samples.

Samples from the Freenome cohort were paired-end sequenced on the Illumina NovaSeq platform.

**Characterization of plasma samples**. Some plasma samples, i.e., of patients B7 and B13[6] and P40, P147, and P148[33] were previously analyzed within other studies. From these analyses, we had information regarding mutations, specific SCNAs, and tumor content of the plasma samples based on the algorithm ichorCNA[39].

*P40*: Mutations: *BRCA1*: NM_007294: Q975R; specific SCNAs: *TMPRSS2-ERG* fusion; *AR* amplification in sample 2; chr12 amplification (containing *ARID2*, *HDAC7*); tumor content: P40_1: 30%; P40_2: 24%;

*P147*: Mutations: *BRCA2*: T298fs; *TP53*: F338I; specific SCNAs: *RET* amplification in sample 3; *AR* amplification; *BRAF* amplification (7q34); *PTEN* loss; tumor content: P147_1: 52%; P147_3: 73%;

*P148*: Mutations: *TP53*: R213X; specific SCNAs: *MYC* amplification; *PTEN* loss; *FOXP1*, *RYBP*, *SHQ1* loss; *TMPRSS2-ERG* fusion; *AR* amplification (lost in P148_3); tumor content: P148_1: 38%; 148_3: 49%.

*C2*: specific SCNAs: high-level amplification on chromosome 12 (*KRAS*) in C2_6, not visible in C2_7; tumor content: C2_6: 18%; C2_7: 28%.

**Selection of TFs**. To select TFs with high-confidence information, we used the 676 TFs contained in the Gene Transcription Regulation Database (GTRD; Version 18.01; http://gtrd.biouml.org)[11], which provides detailed TFBS information based on ChIP-seq data for a variety of tissue samples. We annotated these TFs with an up-to-date curated list of 1639 known or likely human TFs (http://humantfs.ccbr.utoronto.ca/; version 1.01)[1]. Because of the potentially high number of TFBSs per TF, we employed stringent criteria and restricted our analyses to TFs with 1000 TFBSs which were supported by the majority of tissue samples in the GTRD. In total, 172 TFs had fewer than 1000 defined sites in the GTRD and were thus omitted, resulting in 504 TFs amenable to analysis (Supplementary Data 1).

**Transcription factor binding site definitions**. The data from the GTRD database were downloaded (http://gtrd.biouml.org/downloads/18.01/human_meta_clusters.interval.gz), and individual BED files per TF were extracted. The position was recalculated by focusing on the reported point where the meta-cluster has the

highest ChIP-seq signal. The top 1000 sites that were supported by most of the analyzed samples were extracted (1000-msTFBSs). All BED files were then converted to hg19 (from original hg38) using the liftOver tool provided by UCSC.

For the Freenome cohort, BED files were left in hg38 coordinates and regions spanning the ENCODE blacklist were removed.

**Transcription factor binding site overlaps**. In order to check whether binding sites of TFs overlap, regions of the binding sites from GTRD were increased by 25, 50, and 100 bp, respectively, on either side using bedtools slop. Subsequently, the extent of overlap was calculated by using bedtools intersect via pybedtools for every transcription factor with every other transcription factor.

**ATAC-seq data analyses hematological samples**. Raw ATAC-seq data were downloaded from hematological cells from the Gene Expression Omnibus (Accession: GSE74912). The full count matrix was divided by sample. Next, all reads of bins that overlap defined binding sites of a respective transcription factor were summed and divided by the number of total reads within this sample.

**ATAC-seq data analyses cancer samples**. Raw ATAC-seq data matrices of TCGA samples were downloaded for colon adenocarcinoma, prostate carcinoma and breast carcinoma samples (https://api.gdc.cancer.gov/data/f0094e76-4a80-4ee1-9ad0-8ffb94aff5f7). Again, data were divided by sample and bins that overlap a specific transcription factor are summed up and divided by the total amount of reads for a specific sample.

**Analysis of entity-specific TFs**. Since raw ATAC-seq counts were only available for bins where at least one peak was identified, the total number of bins available for analyses diverged between hematological samples and cancer samples. Hence, transcription factor accessibilities from the former analyses were again normalized by the mean accessibility values for that class to correct for that. Finally, for every tumor entity (colon, prostate, and breast carcinoma), transcription factor accessibilities were compared with hematological samples by two-tailed Mann–Whitney U tests, mean ratios, and effect sizes (by Cohen's D).

**Sequencing data preparation**. In order to enhance the nucleosome signal, reads were trimmed to remove parts of the sequencing read that are associated with the linker region. Hence, forward reads were trimmed to only contain base 53–113 (this would correspond to the central 60 bp of a 166 bp fragment). Reads were then aligned to the human hg19 genome using bwa and PCR duplicates were removed using samtools rmdup. Average coverage is calculated by bedtools genomecov.

For the Freenome cohort, reads were aligned to the human hg38 genome using BWA-MEM 0.7.15. Midpoints of paired-end reads were extracted and average midpoint counts at transcription factor binding sites (+/−1250 bp) were calculated per transcription factor. In order to better compare the data to the aforementioned samples, a running median (window size = 30) was applied to the data using the numpy.convolve function in a single dimension.

**MNase-seq data preparation**. BAM files of MNase-seq experiments of GM12878 were downloaded from the ENCODE portal. Reads in BAM files were trimmed directly from the BAM file using pysam. In brief, left-most alignment positions in the BAM file were shifted 53 bp in the respective direction, and the sequence length was adjusted to 60 bp. The coverage patterns were then calculated in the same way as the trimmed cfDNA sequencing data.

**Coverage patterns at transcription factor binding sites**. For every transcription factor in the GTRD, coverage patterns were calculated. To this end, coverage data were extracted for every region using pysam count_coverage in a region ±1000 bp around the defined binding sites. Coverage data at every site were normalized by regional copy-number variation and by mean coverage. For every position around the TFBS, coverage was averaged, and 95% confidence intervals were calculated.

**Measurement of transcription factor accessibility using Savitzky–Golay filters**. As we hypothesized that two distinct signals make up the coverage pattern, two signals of different frequencies were extracted. The lower range frequency data were extracted by a Savitzky–Golay filter (third-order polynomial and window size of 1001). A high-frequency signal was extracted by a different Savitzky–Golay filter (third-order polynomial and window size of 51). The high-frequency signal was then normalized by division by the results of the low-frequency signal. Subsequently, the data range of the high-frequency signal was recorded. Since coverage profiles from TFs with few described binding sites are inherently noisier, a LOESS smoothing was performed over the signal range and the amount of described binding sites. The range values were corrected by the smoothed LOESS, and ranks of the adjusted range were calculated.

**Comparing tumor and control samples**. In order to compare tumor and control samples, the ranks of the respective TFs in the adjusted range values were compared. Rank differences were calculated between a tumor sample and every control sample,

and mean rank differences were recorded. Moreover, z-scores were calculated for every transcription factor from the accessibility ranks by taking the respective rank and subtracting the mean rank of the control samples and dividing by the standard deviation of the transcription factor ranks of the control samples (RankDiff z-scores). In another round, z-scores were calculated for the overall deviation of transcription factor accessibility in two samples. To this end, accessibility values of each of the 24 healthy samples were compared with the remaining 23, and the rank difference was recorded. The rank differences were used to estimate a normal distribution, and z-scores for the rank differences over all TFs were calculated (overall z-scores).

**Analysis of paired samples**. In order to compare subsequent samples from the same patient, we performed pairwise comparisons of every combination of the 24 healthy samples in order to estimate the variability of paired comparisons. The mean and standard deviation of these differences were then used to calculate z-scores of accessibilities of paired samples. Briefly, rank differences between the paired samples were calculated and z-scores of this difference were calculated.

**DNase hypersensitivity data analysis**. BAM_files from DNase hypersensitivity experiments were downloaded from the ENCODE database for GM12878, LNCaP, and HCT116 cell lines. Binding site regions of a transcription factor were increased by 25 bp on either side using bedtools slop. Coverage at the respective binding sites was extracted using mosdepth and normalized by million mapped reads per sample.

**Analysis of somatic copy-number alterations (SCNAs)**. For control data, paired-end alignments were subsampled using samtools view to only include 2% of the initial alignments and converted to FastQ using samtools fastq. For the cancer samples, separate low-coverage WGS was performed. Plasma-Seq[31] was applied to the subsampled FastQ files and the low-coverage data of the cancer samples, respectively. In brief, reads were aligned to the human hg19 genome, and reads were counted within pre-specified bins. The bin size was determined by the amount of theoretically mappable positions to account for differences in mappability throughout the genome. Read counts were normalized for total amount of reads, and GC content of bins were corrected for by LOESS smoothing over the GC spectrum. Moreover, corrected read counts were normalized by the mean read counts of noncancer controls per bin to control for additional positional variation.

**Allele faction threshold estimation**. Accessibilities in colon cancer samples were compared with healthy samples for TFs where evidence of tumor-specific increases and decreases in accessibility was observed in former analyses. Two-tailed Mann–Whitney U tests were performed to compare healthy and cancer samples, as well as subsamples of the cancer data set, based on the respective tumor fraction, as estimated by ichorCNA[39].

**Logistic regression**. In order to classify samples, logistic regression was applied to the accessibility values of all 504 TFs in 971 samples (373 control samples and 598 samples of cancer patients). To this end, the LogisticRegressionCV from the scikit-learn package was applied using fivefold cross-validation and balanced class weights to correct for the slightly unbalanced sample set. In 100 permutations, samples were split into training and test sets. Training samples were used to establish the model whereas (held-out) test sets were used to estimate the actual performance of the model. Mean performance metrics, as well as ROC curves (based on the prediction probabilities of the LogisticRegression) of the models were calculated from the 100 permutations.

**Pairwise comparison of plasma samples**. To address the question whether TF accessibility remains stable over time, we also analyzed two samples each from patients P40, P147, and C2. However, with our very stringent criteria, we did not observe significant differences in these plasma sample pairs (Supplementary Fig. 6).

Between P147_1 and P147_3, a novel, high-amplitude amplification including the *RET* gene evolved, whereas C2_7 had lost an amplification including *KRAS*, which we had observed in the previous sample C2_6. *RET* in prostate cancer and *KRAS* in CRC may affect the PI3K/AKT/mTOR pathway[44], and we therefore investigated downstream targets such as the TF CREB. However, the accessibility was not different from the control plasma samples, and furthermore remained unchanged. Between P40_1 and P40_2, resistance against androgen deprivation therapy (ADT) had evolved, which was reflected in a high-level amplification of the AR gene[45]. However, if AR expanded its repertoire of transcriptional targets, it did not become apparent (Supplementary Fig. 6). Our very conservative approach limiting the analyses to TFs with 1000 TFBSs may explain why we may not have observed differences between these samples.

**Determination of NSE and PSA**. Neurospecific enolase was measured by a commercially available automated sandwich-based ELISA test (Elecsys 2010, Roche, Germany) with a range from LLD of 0.050 to ULD of 370 ng/ml and a reference range from 15.7–17 ng/ml. Inter- and intraassay variances were described with 3.9 and 2.5%, respectively.

PSA was determined by the commercial ARCHITECT Total PSA Assay (Abbot G47859R06), according to the manufacturer's instructions.

**Reporting summary**. Further information on research design is available in the Nature Research Reporting Summary linked to this article.

## Data availability

ATAC-sequencing count matrices of tumors are available from the Genomics Data Commons database [https://api.gdc.cancer.gov/data/47ae33ac-e7ed-488e-88c6-335deccd8712] and ATAC-seq data of hematological cells can be found at the Gene Expression Omnibus with the following accession: GSE74912. Shotgun sequencing data of cfDNA samples pertaining to individual (late-stage) cancer samples, cfDNA pools and 24 controls are available from the European genome-phenome archive database under the accession codes EGAS00001003206. BAM files from DNase hypersensitivity experiments were downloaded from the ENCODE database for GM12878 under the following accessions: ENCFF775ZJX, ENCFF783ZLL, LNCaP under the following accessions: ENCFF002PZG, ENCFF016VTV and HCT116 cell lines under the following accessions: ENCFF081DDV, ENCFF291HHS [https://www.encodeproject.org/search/?type=Experiment&assay_title=DNase-seq]. Sequencing data from control and patient-derived cfDNA from the Freenome cohort are available in Zenodo under the https://doi.org/10.5281/zenodo.2557515. Use of this data is restricted to academic users. All the other data supporting the findings of this study are available within the article and its supplementary information files and from the corresponding author upon reasonable request. A reporting summary for this article is available as a Supplementary Information file.

## Code availability

Code is available in GitHub at https://github.com/PeterUlz/TranscriptionFactorProfiling.

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

## Acknowledgements

The authors gratefully acknowledge Dr. Andrew Godwin (University of Kansas Medical Center) and the University of Kansas Cancer Center's Biospecimen Repository Core Facility staff, funded in part by the National Cancer Institute Cancer Center Support Grant P30 CA168524, National Health Services Research Scotland, Tayside Bior-epository, Geneticist Inc., iSpecimen Inc., and Indivumed for support of this research by providing de-identified plasma samples. This work was supported by CANCER-ID, a project funded by the Innovative Medicines Joint Undertaking (IMI JU; #115749-1), by the BioTechMed-Graz flagship project EPIAge, by the Christian Doppler Research Fund

for Liquid Biopsies for Early Detection of Cancer, and by the Austrian Federal Ministry for Digital and Economic Affairs.

## Author contributions

P.U. and M.R.S. designed the study, S.P., T.M., Q.Z., and J.B. performed the experiments. P.U., I.L., J.B.G., B.W., J.S.J., M.G.S., E.H., and M.R.S. analyzed the data. A.W., A.Z., A.G., G.P., E.P., T.B., and H.S. provided clinical samples and clinical information. B.W., J.S.J., and C.E.S.R. provided access to the Freenome colon cancer cohort. P.U., E.H., C.B., and M.R.S. supervised the study and wrote the paper. All authors revised the paper.

## Competing interests

A patent application has been filed for aspects of the paper (inventors: P.U.; E.H., M.R. S.). E.H. and M.R.S. have an unrelated sponsored research agreement with Servier within CANCER-ID, a project funded by the Innovative Medicines Joint Undertaking (IMI JU), the salary of J.B. was paid through this arrangement. B.W., J.S.J., and C.E.S.R. are employed by Freenome. P.U. was recently hired by Freenome. The remaining authors declare no competing interests.
