## [Peer Review File · Nature Communications]

Reviewers' comments:

Reviewer #1 (Remarks to the Author):

The authors devised a method to trace the origin of cfDNA back to its originating tissue based on an indirect readout of TF binding sites (TFBS) of individual transcription factors. The indirect readout is the positioning of nucleosomes adjacent to a particular TFBS, when average across a thousand instances of a particular TFBS. The manuscript attempts to address an important problem that builds on prior work by Snyder et al. However, the weak magnitude of effects, and the seemingly smorgasbord approach, along with the difficulty in reading the manuscript lead to diminish enthusiasm.

Main issues

1. In general, the use of a large number of TFBS comparisons, the weak (unconvincing) and variable magnitude of effect using an ensemble approach, along with anecdotal examination of a relatively small number of samples from a wide variety of cancers leads one to wonder how broad or selective this study is.
2. Most importantly, to claim the abilities of their method to classify the tumors of origin of cfDNA, they need to classify a series of samples blindly and show them to be different. This would be best done on a wide range of different types of cancer and subtypes of each.
3. Ensemble approaches to site averaging leads one to wonder how much or little an individual site is contributing to the overall effect. A heatmap for each composite/ensemble figure would be helpful.
4. Overall the figures suffer from poor display: micro-fonts, and indistinguishable graph lines, poor labeling of cancer types, that it was extremely tedious to review.
5. Fig. 1e: Except for P148_1, the results are not very compelling.
6. On page 4 starting with "For further confirmation...".. This section is overly brief to the point of being barely understandable.
7. The data in Supp Fig. 5 is hardly convincing.
8. Increased and decreased accessibility as reported by normalized read depth is more convincing when the regions distal to the TFBS are at the same levels as the control. However, many conclusions are based on cases where distal regions deviate as much from the controls as regions close to the TFBS. The basis for this is not explained, but could be due to changes in accessibility in the far-reaches of the genome (maybe heterochromatin), which might lead to altered normalization. For this reason, these results are not particularly convincing.
9. The conclusion related to transdifferentiation of prostate adenocarcinoma to t-SCNC (Table 8) was not convincing. First, the analysis is a one-off event, with no replicates. Second, with >500 different TF_profiles, one would not be surprised that there are some "significant" TFBS changes. Was there correction for multiple hypothesis testing? Please provide references for the TFs and the process indicated in Fig. 3a. Other types of "replicates" or related experiments seemed to be through an ensemble approach, due to sparse data. How is this clinically useful?
10. The statement on page 8 that a unique feature of this approach is that it relies on endogenous DNA fragmentation rather than in vitro fragmentation is an unsupported statement. Endogenous processes might be as, or more, variable than in vitro processes. There is no evidence for or against this statement and so should be removed.
11. The value for clinical diagnostics is not even remotely demonstrated, and over-stated throughout the manuscript. No clinical value is demonstrated.
12. Methods need to be compared directly to that of Snyder et al. Is the Snyder et al pattern already based on a few TFs key to that cancer?
13. Can PCA of cancer samples for figure 3 be done to show clusters by cancer type / subtype?
14. Are the conclusion impacted by the treatment?

Minor Concerns:

15. Please explain figure 2e-2f in more detail. This data is critical to the claim that the method of TF accessibility is appropriate.
16. Supplementary figure 1: X-axis label: Chromosome; Y-axis labels: patient ID; subtypes for example in P147 should be P147_1 and P147_3
17. Corresponding text for DHS coverage of samples (pg 4) for top of panel 1c makes no sense. "resulting for the cancer-derived samples in decreased amplitudes"

18. Figures S2b and S2c are based on CTCF, why not just show coverage profile of CTCF rather than 2 other TFs
19. Supp. Figure 4 Legends within each individual figure are too small to read
20. S4b: "dependent on whether all tissues in the GTRD or whether, more strictly, only those peaks that are supported by >50% of the maximum number of samples (>50%-TFBSs) were included to "dependent on whether all tissues in the GTRD (left) or whether, more strictly, only those peaks that are supported by >50% of the maximum number of samples (>50%-TFBSs) were included (right)"; also this distinction should be made with a title along the upper X-axis for this panel.
21. S4e right boxplot, y-axis should be "Percentage overlap with TSSs" not "Percentage overlap with CpG Islands"
22. Is this total, free, or complexed PSA? This would matter in terms of an adenocarcinoma
23. No method is described, corresponding to how PSA or NSE were determined

Reviewer #2 (Remarks to the Author):

Title: Inference of tumor cell-specific transcription factor binding from cell-free DNA
 Authors: Ulz, P et al

Summary: This study reports the analysis of cell-free DNA from cancer patients and individuals without cancer to infer the activity of transcription factors in tumor cells using nucleosome maps. The inference is based on high read counts from low pass whole genome sequencing of cfDNA in the midpoint of a canonical nucleosome, indicating a region of gene activity (e.g. transcription). The methods were used to assess the activity of specific transcription factors active in prostate cancer such as the AR, HOXB13 and NKX3-1. Notably, the inferred transcription factor activity changed over a treatment continuum with the emergence of therapy resistance via the activation of a transdifferentiation program of neuroendocrine phenotype.

Comments:

1. This manuscript extends previous work by this group where maps of nucleosome occupancy at transcription start sites were generated from cfDNA. Consequently, the novelty/innovation of the present study is somewhat diminished. Further, the work also extends previous studies demonstrating that cfDNA analyses can be used to inform tissues of origin (Snyder et al). Both studies are acknowledged in the present manuscript.
2. The overall concept here is very interesting and also has the potential to impact clinical medicine. Notably, it extends the use of cfDNA/ctDNA beyond simply assessing somatic genomic alterations such as mutation or structural aberrations by providing information capable of classifying cell phenotype.
3. Overall, the manuscript is very well written. The objectives are clear, the figures are informative, and sufficient detail is provided in the methods to allow reproduction.
4. The major innovation centers on developing approaches to define the activity of individual transcription factors (TFs) that may contribute to cell identify and also potentially to drivers and phenotypes of neoplasia (e.g. drug resistance).
5. It would be useful for the authors to explicitly state the major distinguishing features of the present study, from their prior work (Ulz et al Nature Genetics 2016).
6. A strength of the study is the use of a large number of patient-derived cfDNA samples, several of which were analyzed over a longitudinal treatment continuum.
7. A minor limitation is the lack of direct analysis of the tumors/tissues from which the cfDNA/ctDNA arose, to directly conform the inferences of transcription factor activity and gene expression.
8. On Pg.5 – it is not clear which samples are used in the benchmarking and comparison?
9. Pg.7 and Supp Fig 5 – the signals (difference in coverage) is weak, especially for NKX3.1. What is the confidence that NKX3.1 has increased accessibility?
10. Pg.7 – for the pooled shallow analysis, what is the effect of variable ctDNA fraction? It would be useful to comment/show the confidence to call accessibility as a function of ctDNA fraction?
11. Pg. 7 and Fig 3e – Comparison between time points using Z-score may be difficult to interpret: First, a z-score assumes that the ranks are normally distributed. Can this be confirmed? Second,

how are the difference in the z-scores between samples interpreted? For example with N-MYC, 4.4 (P148_3) vs 2.6 (P148_1) has a subtle difference of 1.8 but other TF examples show larger differences. Is there a significance or confidence measure to help establish a true change in accessibility? This measure should also account for the individual sample variability.

12. Pg. 8 and Fig 3f – Similar to comment above, what is an appropriate z-score threshold to determine the accessibility status?

Reviewer #3 (Remarks to the Author):

The work of Ulz et al. builds on a previous work from the Speicher group where they developed the basis for inferring gene expression in cells of origin from deep sequencing of cfDNA, by analysis of nucleosome occupancy patterns.

In the current manuscript the authors devised a modified computational pipeline to assess TF activity from cfDNA sequencing. The idea is to investigate nucleosome positioning patterns for transcription factor binding sites (i.e. fewer cfDNA sequence reads from TF binding sites). Since TFs have many binding sites this has the potential to increase detection sensitivity. By applying the pipeline to generate meta plots on such TF-regulated gene sets the authors extract information regarding TF activity, potentially with low sequencing coverage and cost.

The authors show that this approach can detect TF activity from cfDNA sequencing, and further show that they can call TFs that show differential nucleosome-occupancy between cfDNA of cancer patients compared to healthy subjects.

This is a clever approach that has the potential to probe transcription factor activity in cancer from cfDNA.

The manuscript is clear and well written and to the best of my judgment the computational methodology is adequate.

The manuscript is of interest to the large scientific communities and is suitable for publication in Nature Communications.

While the results presented here are reasonably convincing it often feels anecdotal in terms of the analysis and the number of cancer samples used.

Addressing the following points will be of importance:

1. Since the authors claim that they can detect TF binding from shallow sequencing data, the manuscript will benefit from analysing more samples to make more diagnostic claims. In fact, the authors can use cfDNA data from other groups to strengthen the manuscript.
2. It is not always clear how the authors chose the TFs they focus on and to what extent these TFs are suitable for cancer diagnosis. For example, in Figure 3 the authors focus on AR, HOXB13, and NKX3-1. Beyond the fact that these factors are connected to prostate cancer, how did the authors select these genes? Are these TFs informative if contrasted against additional cell types?
3. The authors claim that their method reports on TFs binding in the cells of origin. They bring some support by showing that the nucleosome patterns they observe are in agreement with some aspects of known biology. It would be important to support this idea with an alternative independent approach. One option is to contrast the cfDNA data with RNA-seq and ATAC-seq from the tumors (at least for a few cases). Another potential alternative is to perform cfDNA-seq without size selection that preserves short fragments (<100bps), which should enrich for sequences of bound TFs (since it is not covered by a nucleosome that generate larger fragments). Such analysis will support the idea that nucleosome patterns are indeed resulting from TF binding.
4. The authors should discuss in more detail the current detection limit of the assay. This can help the community to understand to what extent (in terms of ctDNA fraction) the method is applicable. One way to do that is by diluting plasma from cancer patients into healthy plasma and repeating the TF analysis. In combination with information on the fraction of ctDNA the authors should be able to adequately discuss detection limits. In any case, please state upfront current sensitivity limits of method (e.g. utility only when the cancer fraction in cfDNA is above X%). It is understood that this is a method in development that is not ready yet for clinical use in real life samples with 0.1% cancer cfDNA, and there is no need to hide current limitations.
5. One aspect that is relevant to all cfDNA epigenetics studies is the claim that it can detect

cancer. It is unfortunate that the authors did not test other pathologies (inflammation vs cancer for example). This point, if not experimentally addressed should at least be discussed.

Reviewer #1:

The authors devised a method to trace the origin of cfDNA back to its originating tissue based on an indirect readout of TF binding sites (TFBS) of individual transcription factors. The indirect readout is the positioning of nucleosomes adjacent to a particular TFBS, when average across a thousand instances of a particular TFBS. The manuscript attempts to address an important problem that builds on prior work by Snyder et al. However, the weak magnitude of effects, and the seemingly smorgasbord approach, along with the difficulty in reading the manuscript lead to diminish enthusiasm.

Our response: We are pleased that the reviewer acknowledges that we are dealing with an important problem in our manuscript. At the same time, the criticism (“weak magnitude of effects”, “smorgasbord approach”, “difficulty in reading the manuscript”) is well taken and we addressed these issues as outlined in the following.

Main issues

1. In general, the use of a large number of TFBS comparisons, the weak (unconvincing) and variable magnitude of effect using an ensemble approach, along with anecdotal examination of a relatively small number of samples from a wide variety of cancers leads one to wonder how broad or selective this study is.

Our response: We are grateful that this reviewer raises this important issue, i.e. the “large number of TFBS comparisons”. The subject was taken up again later in point 9 (“...with >500 different TF_profiles, one would not be surprised that there are some “significant” TFBS changes”). In fact, we completely agree with this reviewer that the selection of TFs/TFBSs is of utmost importance and the selection process was indeed not described well in the first draft of the manuscript. To address this highly relevant point, we decided on the following strategy:

we extracted tissue and cancer type-specific peak sets from recent large-scale studies where TF binding sites were assessed in hematopoietic lineages and tumor samples from The Cancer Genome Atlas (TCGA) using ATAC-seq (Corces et al. [2016] Nat Genet 48:1193-1203; Corces et al. [2018] Science 362:eaav1898). As a result of these efforts, we generated comprehensive lists of high-confidence TFs and their respective accessibilities in hematopoietic lineages and three different epithelial lineages (i.e. breast, prostate, colon). This approach allowed us then to compare the accessibilities of the TFBSs as defined by ATAC-seq with those observed by us in the plasma DNA of patients with the respective epithelial malignancies. As outlined in the new version of the manuscript, we found excellent agreements.

Furthermore, we regret the presentation of our results in the first draft of the manuscript, which caused this reviewer to describe the observed effects as weak and unconvincing. In fact, we revised the entire statistical evaluation and description and accordingly also the presentation of the data. In the new version of the manuscript, we have considerably extended the explanation of the used z-score statistics. In the literature, z-score statistic thresholds are usually set to ± 3 . To be more stringent and to demonstrate that we in fact do not observe weak effects, we raised the thresholds to ± 5 . We hope that with this aspect together with the newly designed figures that it now becomes obvious that the observed changes of TFBS accessibilities in plasma DNA are in fact strong effects.

As this reviewer also strongly critiqued the readability of the manuscript, we carefully rewrote the entire manuscript. We considerably streamlined the text and we believe that we succeeded in creating a much clearer presentation of our data.

2. Most importantly, to claim the abilities of their method to classify the tumors of origin of cfDNA, they need to classify a series of samples blindly and show them to be different. This would be best done on a wide range of different types of cancer and subtypes of each.

Our response: This point is well taken. However, in the aforementioned seminal paper, Corces and colleagues analyzed 410 tumor samples from 23 types of primary human cancers by ATAC-seq (Corces et al. [2018] Science 362:eaav1898). Identification of cancer subtypes from ATAC-seq data could be performed for only three cancer types with a sufficient number of available donors (i.e. breast: $n=74$; prostate: $n=26$; kidney: $n=34$). Importantly, the authors discuss their study as “an initial characterization of the chromatin regulatory landscape in human cancers” and they stated: “as the chromatin accessibility landscapes of additional primary cancer samples are profiled, we anticipate the identification of further epigenetic subdivisions with prognostic implications, potentially nominating avenues for therapeutic intervention.” Hence, identification of cancer types and in particular of subtypes based on accessibility landscapes is an emerging field, which is at present in its infancy and sufficient datasets are not available yet. The purpose of our study was to develop novel tools to assess these accessibilities in plasma DNA, which we realized for a number of high-interest TFs. Furthermore, for cancer subtyping, which is of high relevance for clinical applications, we focused as a proof-of-concept on prostate cancer cases in the identification of adenocarcinoma vs. neuroendocrine carcinoma, as there is a large body of literature about different accessibilities of TFBSs in these prostate cancer subtypes (respective references are listed in our response to point 9). As a result, we show, to the best of our knowledge for the first time, that cancer subtyping based on TFBS accessibility may be feasible from liquid biopsy. We are convinced that as more chromatin accessibility landscapes in human cancers are established –as suggested by Corces and colleagues- that this information can then be transferred and applied non-invasively to a plasma-based approach as described in our manuscript.

3. Ensemble approaches to site averaging leads one to wonder how much or little an individual site is contributing to the overall effect. A heatmap for each composite/ensemble figure would be helpful.

Our response: We agree with this reviewer and we extensively changed the figures. We appreciate the idea with a heatmap, but we favored instead the use of plots illustrating the overall z-score vs. the accessibility, as these impressively show –in our opinion- the altered TFBS accessibilities in our plasma samples.

4. Overall the figures suffer from poor display: micro-fonts, and indistinguishable graph lines, poor labeling of cancer types, that it was extremely tedious to review.

Our response: We regret the poor quality of the figures and we took care that the revised versions of the figures are without the limitations raised by this reviewer.

5. Fig. 1e: Except for P148_1, the results are not very compelling.

Our response: As mentioned above, all illustrations and representations of the data have been carefully revised. We are grateful to this reviewer, as his/her justified criticism initiated a complete overhaul of our statistical analyses, inclusion of ATAC-seq data, revision of figures, and rewriting of the manuscript.

6. On page 4 starting with “For further confirmation...”.. This section is overly brief to the point of being barely understandable.

Our response: We agree with the reviewer and in the new version of the manuscript, we omitted several of the confirmatory efforts, which were summarized in this section. To improve the readability of the manuscript, we decided that some of the confirmatory analyses were not relevant but rather confusing. In the current version of the manuscript, we now confine to two additional confirmatory data evaluations, i.e. a comparison of our results with ENCODE data and the use of high molecular weight DNA as a negative control. We hope that this part of the manuscript is now more comprehensible.

7. The data in Supp Fig. 5 is hardly convincing.

Our response: We agree and as mentioned above, all illustrations and data representations were carefully revised and some figures, such as Supp. Fig. 5, were completely exchanged.

8. Increased and decreased accessibility as reported by normalized read depth is more convincing when the regions distal to the TFBS are at the same levels as the control. However, many conclusions are based on cases where distal regions deviate as much from the controls as regions close to the TFBS. The basis for this is not explained, but could be due to changes in accessibility in the far-reaches of the genome (maybe heterochromatin), which might lead to altered normalization. For this reason, these results are not particularly convincing.

Our response: We took care that normalization is not based on the far-reaches of the genome. In fact, during the calculation of accessibility values, coverage values are normalized by the mean coverage in the entire 2,000bp window. Thus, even if normalized coverage plots appear not to start or end at a relative coverage of 1, this has been factored in in the analysis. Moreover, heterochromatin/euchromatin rarely changes within 2,000bp since these building blocks of chromosomes far exceed the analyzed genomic size that we are looking at. Based on this, individual TFBSs (supported also by experimental ChIP-seq evidence) and the flanking $\pm 1,000$ bp regions both are in euchromatin.

9. The conclusion related to transdifferentiation of prostate adenocarcinoma to t-SCNC (Table 8) was not convincing. First, the analysis is a one-off event, with no replicates. Second, with >500 different TF_profiles, one would not be surprised that there are some “significant” TFBS changes. Was there correction for multiple hypothesis testing? Please provide references for the TFs and the process indicated in Fig. 3a. Other types of “replicates” or related experiments seemed to be through an ensemble approach, due to sparse data. How is this clinically useful?

Our response: The points raised by this reviewer are well taken. It is correct that we have indeed only one case with serial plasma samples collected before and after transdifferentiation of the prostate cancer from an adenocarcinoma to a neuroendocrine tumor. However, this is to the best of our knowledge the very first example of significant changes in TFBS accessibility during the course of a cancer disease established non-invasively from peripheral blood. In order to avoid the description of a “one-off event”, we added plasma samples from four further patients with proven t-SCNC to show that TFBS accessibility differs in prostate cancer subtypes. As shown now in the newly designed Figure 3d, TFBS accessibility is clearly significantly different in adenocarcinoma and neuroendocrine carcinoma. Due to the current limited knowledge about TF accessibility in cancers and their subtypes (see the respective discussion from the Corces et al. paper in point 2), we think that this prostate cancer scenario is particularly apt for our purposes as detailed descriptions about TF changes in neuroendocrine prostate cancer has been extensively described, e.g. in the following references, which are all cited in the revised manuscript: Puca et al. [2018] Cold Spring Harbor Perspectives in Medicine a030593; Svensson et al. [2014] Nucleic Acids Res 42:999-1015; Puca et al. [2018] Nat Commun 9:2404, and Beltran et al. [2016] Nat Med 22:298-305. The clinical utility of identifying the transdifferentiation process are two-fold: First, the transdifferentiation of an adenocarcinoma to a t-SCNC is a frequent (~20%) mechanism in the development of treatment resistance and therefore of high relevance in the most frequent tumor entity of men. Second, identification of this process has clinically significant implications because it requires a change in therapy (Regarding clinical utility please see ref. 32, i.e. Aggarwal et al. [2018] J Clin Oncol 36:2492-2503).

10. The statement on page 8 that a unique feature of this approach is that it relies on endogenous DNA fragmentation rather than in vitro fragmentation is an unsupported statement. Endogenous processes might be as, or more, variable than in vitro processes. There is no evidence for or against this statement and so should be removed.

Our response: We agree with this reviewer and we deleted this statement.

11. The value for clinical diagnostics is not even remotely demonstrated, and over-stated throughout the manuscript. No clinical value is demonstrated.

Our response: As an example for clinical diagnostics, we demonstrated tumor subtype prediction in prostate carcinoma. Furthermore, we are convinced that when more advanced open chromatin accessibility landscapes become available (see the discussion above of the Corces et al. data), the value of our plasma DNA analysis strategy for clinical diagnostics will increase. In the revised version of the manuscript, we took care to avoid over-stating the clinical value of our approach and we rephrased in a more careful manner in the respective parts of the manuscript.

12. Methods need to be compared directly to that of Snyder et al. Is the Snyder et al pattern already based on a few TFs key to that cancer?

Our response: We significantly extended the discussion and provide now a detailed comparison with the Snyder et al. study. We added the following to the discussion:

“Compared to a previous publication by Snyder and colleagues (ref. 5), our study has several distinct differences. First, Snyder and colleagues did not profile individual TFs but instead established general tissue-specific patterns using mixtures of cfDNA signals resulting from multiple cell types and analyses by Fourier transformation (ref. 5). In contrast, we utilized recently generated ATAC-seq data (ref. 12, 13) and profiled individual TFs and thereby established lineage-specific TFs for clinical applications. Second, we introduce a novel metric, the accessibility score, which, by enabling the objective comparison of TF binding events in various plasma samples, paves the way for entirely new diagnostic procedures. Third, we were for the first time able to use cfDNA to show TFBS plasticity during a disease course, such as reprogramming to a different cell lineage in prostate cancer. Such a dynamic view of TF activity (vis-à-vis the static view obtained from tissue (ref. 13)) is a unique feature of cfDNA

analyses. Fourth, we demonstrate that our cfDNA TFBS bioinformatics pipeline allows subclassification of tumor entities and hence may address an important diagnostic dilemma in the managing of patients with prostate cancer (ref. 32). To the best of our knowledge, this is the first time that tumor subtype prediction from liquid biopsies and in particular using transcription factors has been shown. Fifth, whereas Snyder and colleagues required 1.5 billion reads per plasma sample (ref. 5), which is prohibitive for routine clinical use from a cost perspective, we were able to conduct in-depth TF analysis with ~50 million reads, making our approach more amenable to clinical applications. Finally, we used a large cohort of patients with colon cancer to establish the resolution limits and provide evidence that our approach is suitable for detection of early-stage cancer. As such, our approach may enable the detection of cancer at stages when the disease is most treatable/curable and thereby help to reduce cancer mortality rates (ref. 40).”

13. Can PCA of cancer samples for figure 3 be done to show clusters by cancer type / subtype?

Our response: We thank the reviewer for this suggestion. However, we revised the entire statistical procedure to demonstrate that we are not describing “weak effects” (see above). Due to this extensive revision, we do not see that principle component analyses could result in a further improvement in the description of observed effects.

14. Are the conclusion impacted by the treatment?

Our response: We thank this reviewer for this interesting question. A well-documented example is the aforementioned transdifferentiation of an adenocarcinoma to a t-SCNC, which has been shown to be a frequent (~20%) mechanism in the development of treatment resistance to androgen deprivation therapy (for a recent publication see ref. 32; i.e. Aggarwal et al. [2018] J Clin Oncol 36:2492-2503). However, in order to answer the question of the impact by treatment correctly, intensive longitudinal studies would be needed to test at a systematic level how certain treatments may affect TFBS accessibility. Such studies are beyond the scope of our manuscript, which aimed at demonstrating that open chromatin regions, in particular TFBSs, are amenable to appropriate plasma DNA analyses.

Minor Concerns:

15. Please explain figure 2e-2f in more detail. This data is critical to the claim that the method of TF accessibility is appropriate.

16. Supplementary figure 1: X-axis label: Chromosome; Y-axis labels: patient ID; subtypes for example in P147 should be P147_1 and P147_3

18. Figures S2b and S2c are based on CTCF, why not just show coverage profile of CTCF rather than 2 other TFs

19. Supp. Figure 4 Legends within each individual figure are too small to read

21. S4e right boxplot, y-axis should be “Percentage overlap with TSSs” not “Percentage overlap with CpG Islands”

Our response: Minor concerns 15, 16, 18, 19, and 21 related to the quality of figures. By redesigning all figures, we took care to address the justified criticism by this reviewer.

17. Corresponding text for DHS coverage of samples (pg 4) for top of panel 1c makes no sense. “resulting for the cancer-derived samples in decreased amplitudes

Our response: In the completely revised version of the manuscript, this statement was corrected.

20. S4b: “dependent on whether all tissues in the GTRD or whether, more strictly, only those peaks that are supported by >50% of the maximum number of samples (>50%-TFBSs) were included to “dependent on whether all tissues in the GTRD (left) or whether, more strictly, only those peaks that are supported by >50% of the maximum number of samples (>50%-TFBSs)

were included (right)”; also this distinction should be made with a title along the upper X-axis for this panel.

Our response: We realized that the use of three different stringency criteria caused a lot of confusion and we apologize for this. As the three stringency criteria did not add significant information to the study, we decided in the new version to confine our analyses to one of these three criteria, i.e. to high confidence 504 TFs where the majority of tissue samples in the Gene Transcription Regulation Database (GTRD) supported 1,000 TFBSs per TF.

22. Is this total, free, or complexed PSA? This would matter in terms of an adenocarcinoma

23. No method is described, corresponding to how PSA or NSE were determined

Our response: We added this information to the manuscript.

Reviewer #2:

Summary: This study reports the analysis of cell-free DNA from cancer patients and individuals without cancer to infer the activity of transcription factors in tumor cells using nucleosome maps. The inference is based on high read counts from low pass whole genome sequencing of cfDNA in the midpoint of a canonical nucleosome, indicating a region of gene activity (e.g. transcription). The methods were used to assess the activity of specific transcription factors active in prostate cancer such as the AR, HOXB13 and NKX3-1. Notably, the inferred transcription factor activity changed over a treatment continuum with the emergence of therapy resistance via the activation of a transdifferentiation program of neuroendocrine phenotype.

Our response: This is a nice summary of the first previous version of our manuscript.

1. This manuscript extends previous work by this group where maps of nucleosome occupancy at transcription start sites were generated from cfDNA. Consequently, the novelty/innovation of the present study is somewhat diminished. Further, the work also extends previous studies demonstrating that cfDNA analyses can be used to inform tissues of origin (Snyder et al). Both studies are acknowledged in the present manuscript.

Our response: As mentioned above (comment 12 by Reviewer #1), we now explain the differences between the work by Snyder et al. and our study in detail and why we are convinced that our study has a high degree of novelty and is very innovative. As already stated above, we added the following to the discussion:

“Compared to a previous publication by Snyder and colleagues (ref. 5), our study has several distinct differences. First, Snyder and colleagues did not profile individual TFs but instead established general tissue-specific patterns using mixtures of cfDNA signals resulting from multiple cell types and analyses by Fourier transformation (ref. 5). In contrast, we utilized recently generated ATAC-seq data (ref. 12, 13) and profiled individual TFs and thereby established lineage-specific TFs for clinical applications. Second, we introduce a novel metric, the accessibility score, which, by enabling the objective comparison of TF binding events in various plasma samples, paves the way for entirely new diagnostic procedures. Third, we were for the first time able to use cfDNA to show TFBS plasticity during a disease course, such as reprogramming to a different cell lineage in prostate cancer. Such a dynamic view of TF activity (vis-à-vis the static view obtained from tissue (ref. 13)) is a unique feature of cfDNA analyses. Fourth, we demonstrate that our cfDNA TFBS bioinformatics pipeline allows subclassification of tumor entities and hence may address an important diagnostic dilemma in the managing of patients with prostate cancer (ref. 32). To the best of our knowledge, this is the first time that tumor subtype prediction from liquid biopsies and in particular using transcription factors has been shown. Fifth, whereas Snyder and colleagues required 1.5 billion reads per plasma sample (ref. 5), which is prohibitive for routine clinical use from a cost perspective, we were able to conduct in-depth TF analysis with ~50 million reads, making our approach more amenable to clinical applications. Finally, we used a large cohort of patients with colon cancer to establish the resolution limits and provide evidence that our approach is suitable for detection of early-stage cancer. As such, our approach may enable the detection

of cancer at stages when the disease is most treatable/curable and thereby help to reduce cancer mortality rates (ref. 40).”

2. The overall concept here is very interesting and also has the potential to impact clinical medicine. Notably, it extends the use of cfDNA/ctDNA beyond simply assessing somatic genomic alterations such as mutation or structural aberrations by providing information capable of classifying cell phenotype.

3. Overall, the manuscript is very well written. The objectives are clear, the figures are informative, and sufficient detail is provided in the methods to allow reproduction.

4. The major innovation centers on developing approaches to define the activity of individual transcription factors (TFs) that may contribute to cell identify and also potentially to drivers and phenotypes of neoplasia (e.g. drug resistance).

6. A strength of the study is the use of a large number of patient-derived cfDNA samples, several of which were analyzed over a longitudinal treatment continuum.

Our response: We highly appreciate the encouraging comments 2, 3, 4 and 6.

5. It would be useful for the authors to explicitly state the major distinguishing features of the present study, from their prior work (Ulz et al Nature Genetics 2016).

Our response: We made the requested changes to the new version of the manuscript.

7. A minor limitation is the lack of direct analysis of the tumors/tissues from which the cfDNA/ctDNA arose, to directly conform the inferences of transcription factor activity and gene expression.

Our response: This point is well taken. We have decided to use publicly available ATAC-seq data for confirmation, as we believe it is more meaningful. As outlined in our response to Reviewer #1, we extracted tissue and cancer type-specific peak sets from recent large-scale studies where TF binding sites were assessed in hematopoietic lineages and tumor samples from The Cancer Genome Atlas (TCGA) using ATAC-seq (Corces et al. [2016] Nat Genet 48:1193-1203; Corces et al. [2018] Science 362:eaav1898). As a result of these efforts, we generated comprehensive lists of high-confidence TFs and their respective accessibilities in hematopoietic lineages and three different epithelial lineages (i.e. breast, prostate, colon). This approach allowed us then to compare the accessibilities of the TFBSs as defined by ATAC-seq with those observed by us in the plasma DNA of patients with the respective epithelial malignancies. In the new manuscript version, we show an extremely high agreement between TF binding site accessibility measured by ATAC-seq and our plasma DNA analysis.

8. On Pg.5 – it is not clear which samples are used in the benchmarking and comparison?

Our response: We rephrased this part of the manuscript.

9. Pg.7 and Supp Fig 5 – the signals (difference in coverage) is weak, especially for NKX3.1. What is the confidence that NKX3.1 has increased accessibility?

Our response: We completely agree with this reviewer. We redid the entire statistical analyses and now use a very stringent z-score statistic with ± 5 thresholds (usually ± 3 thresholds are employed) and furthermore all figures were carefully revised.

10. Pg.7 – for the pooled shallow analysis, what is the effect of variable ctDNA fraction? It would be useful to comment/show the confidence to call accessibility as a function of ctDNA fraction?

Our response: The individual low-coverage pooled cfDNA samples do not allow us to do this kind of analyses, since nucleosome positioning is very noisy for these samples. However, we newly added to the manuscript the analyses of an additional cohort consisting of 592 plasma samples from individuals with colon cancer and for each of these plasma samples we calculated the tumor fraction using the ichorCNA algorithm (Adalsteinsson et al. [2017] Nat

Commun 8:1324). With this dataset we show accessibility in relation to tumor fraction for six transcription factors, which is depicted in the new Figure 4a.

11. Pg. 7 and Fig 3e – Comparison between time points using Z-score may be difficult to interpret: First, a z-score assumes that the ranks are normally distributed. Can this be confirmed? Second, how are the difference in the z-scores between samples interpreted? For example with N-MYC, 4.4 (P148_3) vs 2.6 (P148_1) has a subtle difference of 1.8 but other TF examples show larger differences. Is there a significance or confidence measure to help establish a true change in accessibility? This measure should also account for the individual sample variability.

Our response: We are grateful for these comments. First, we tested whether the z-score ranks are normally distributed and found that this was indeed the case (illustrated in the new Supplementary Figure 4). Second, we appreciate the comments regarding the differences in the z-scores between samples interpreted. In the revised version we now illustrate the z-scores of the two serial plasma samples (i.e. P148_1 [adenocarcinoma] and P148_3 [neuroendocrine carcinoma]) next to each other to display the tremendous differences (new Fig. 3a, left panel). In addition, we also calculated z-scores in a pairwise comparison between P148_1 and P148_3. Whereas accessibility of hematopoietic TFs remained unchanged as expected, GRHL2 and NKX3-1 exceeded the -3 z-score and importantly AR, HOXB13, and REST exceeded the -5 z-score (new Fig. 3a, right panel). Hence, with our revised statistical approach, we show that we can establish true changes in accessibility and furthermore that tremendous differences in TFBS accessibility may indeed happen during a disease course.

12. Pg. 8 and Fig 3f – Similar to comment above, what is an appropriate z-score threshold to determine the accessibility status?

Our response: Our revised statistical analyses are based on ± 5 z-score thresholds. In our opinion, this is a very stringent threshold (usual a ± 3 threshold in z-score statistics is employed), which reveals differences in TFBS accessibility status in a very reliable way.

Reviewer #3:

The work of Ulz et al. builds on a previous work from the Speicher group where they developed the basis for inferring gene expression in cells of origin from deep sequencing of cfDNA, by analysis of nucleosome occupancy patterns.

In the current manuscript the authors devised a modified computational pipeline to assess TF activity from cfDNA sequencing. The idea is to investigate nucleosome positioning patterns for transcription factor binding sites (i.e. fewer cfDNA sequence reads from TF binding sites). Since TFs have many binding sites this has the potential to increase detection sensitivity. By applying the pipeline to generate meta plots on such TF-regulated gene sets the authors extract information regarding TF activity, potentially with low sequencing coverage and cost.

The authors show that this approach can detect TF activity from cfDNA sequencing, and further show that they can call TFs that show differential nucleosome-occupancy between cfDNA of cancer patients compared to healthy subjects.

This is a clever approach that has the potential to probe transcription factor activity in cancer from cfDNA.

The manuscript is clear and well written and to the best of my judgment the computational methodology is adequate.

The manuscript is of interest to the large scientific communities and is suitable for publication in Nature Communications.

Our response: We were very happy about the evaluation of our manuscript by this reviewer.

While the results presented here are reasonably convincing it often feels anecdotal in terms of the analysis and the number of cancer samples used.

Addressing the following points will be of importance:

1. Since the authors claim that they can detect TF binding from shallow sequencing data, the manuscript will benefit from analysing more samples to make more diagnostic claims. In fact, the authors can use cfDNA data from other groups to strengthen the manuscript.

Our response: This point is well taken. However, we used shallow-coverage (<0.2x) data only as pools of multiple cfDNA samples. In fact, we show that we need at least ~50 million reads to conduct our TFBS accessibility analyses and there are not too many samples with these characteristics in the publicly accessible cfDNA pool.

However, to address this point and also point 4 by this reviewer, we newly added to the manuscript the analyses of an additional cohort consisting of 592 plasma samples from individuals with colon cancer. For each of these plasma samples, we calculated the tumor fraction using the ichorCNA algorithm (Adalsteinsson et al. [2017] Nat Commun 8:1324), which allowed us to show TF accessibility in relation to tumor fraction for several transcription factors (depicted in the new Figure 4a).

2. It is not always clear how the authors chose the TFs they focus on and to what extent these TFs are suitable for cancer diagnosis. For example, in Figure 3 the authors focus on AR, HOXB13, and NKX3-1. Beyond the fact that these factors are connected to prostate cancer, how did the authors select these genes? Are these TFs informative if contrasted against additional cell types?

Our response: We agree with the reviewer that in the first version of the manuscript we did indeed not describe well as to how we chose the TFs we focused on (similar concerns were raised by reviewer #1). As outlined in one of our responses to reviewer #1, we now addressed this highly relevant point with the following strategy: we extracted tissue and cancer type-specific peak sets from recent large-scale studies where TF binding sites were assessed in hematopoietic lineages and tumor samples from The Cancer Genome Atlas (TCGA) using ATAC-seq (Corces et al. [2016] Nat Genet 48:1193-1203; Corces et al. [2018] Science 362:eaav1898). As a result of these efforts, we generated comprehensive lists of high-confidence TFs and their respective accessibilities in hematopoietic lineages and three different epithelial lineages (i.e. breast, prostate, colon). This approach allowed us then to compare the accessibilities of the TFBSs as defined by ATAC-seq with those observed by us in the plasma DNA of patients with the respective epithelial malignancies. In other words, we knew from the publicly available ATAC-seq data which TFBSs have an increased accessibility in tissue and we compared the accessibility in plasma for those TFs and found –as outlined in the new version of the manuscript- excellent agreements.

3. The authors claim that their method reports on TFs binding in the cells of origin. They bring some support by showing that the nucleosome patterns they observe are in agreement with some aspects of known biology. It would be important to support this idea with an alternative independent approach. One option is to contrast the cfDNA data with RNA-seq and ATAC-seq from the tumors (at least for a few cases). Another potential alternative is to perform cfDNA-seq without size selection that preserves short fragments (<100bps), which should enrich for sequences of bound TFs (since it is not covered by a nucleosome that generate larger fragments). Such analysis will support the idea that nucleosome patterns are indeed resulting from TF binding.

Our response: As outlined in the previous point, we conducted a comprehensive analysis of publicly available ATAC-seq data from tissue and blood and found an excellent agreement with our nucleosome patterns from plasma DNA. For this reason, we are convinced that inclusion of the ATAC-seq data demonstrate that the nucleosome patterns, which we observed in plasma DNA, indeed result from TF binding.

4. The authors should discuss in more detail the current detection limit of the assay. This can help the community to understand to what extent (in terms of ctDNA fraction) the method is applicable. One way to do that is by diluting plasma from cancer patients into healthy plasma and repeating the TF analysis. In combination with information on the fraction of ctDNA the

authors should be able to adequately discuss detection limits. In any case, please state upfront current sensitivity limits of method (e.g. utility only when the cancer fraction in cfDNA is above X%). It is understood that this is a method in development that is not ready yet for clinical use in real life samples with 0.1% cancer cfDNA, and there is no need to hide current limitations.

Our response: We highly appreciate this issue of the detection limit of the assay, in particular as detection of early-stage cancer represents a very current topic in the liquid biopsy community. To address this important issue accurately, we added the analyses of the aforementioned additional cohort consisting of 592 plasma samples from individuals with colon cancer, where 82.1% of patients had stage I ($n=197$) or stage II ($n=280$) disease and compared those to 177 plasma samples from subjects with no current cancer diagnosis (for this newly added cohort the mean read count per sample was ~299.2 million, standard deviation 100.4 million reads). As outlined in the revised version of the manuscript we found that our approach is capable of identifying ctDNA in patients with stage I colon cancer with a 74% precision (95% CI: 0.53-0.90), 71% sensitivity (recall) (95% CI: 0.54-0.87), and 72% specificity (0.48-0.90, 95% CI), and in patients with stage II colon cancer with a 84% precision (95% CI: 0.69-1.0), 74% sensitivity (recall) (95% CI: 0.58-0.88), and 77% specificity (0.56-1.0, 95% CI) (The results are displayed in the new Figure 4). In the Discussion, we then compare our approach with other recently published liquid biopsy based methods aiming at early cancer detection.

5. One aspect that is relevant to all cfDNA epigenetics studies is the claim that it can detect cancer. It is unfortunate that the authors did not test other pathologies (inflammation vs cancer for example). This point, if not experimentally addressed should at least be discussed.

Our response: We agree with this reviewer and we added this highly relevant point to the discussion.

In summary, the main critical new information or data, which we added to the new version of the manuscript, include:

- **Improved selection of high-confidence transcription factors (TF):** As one concern was the selection of TFs/TFBSs, and as we agree that the selection process was indeed not described well in the first draft of the manuscript, we added the following information: we extracted tissue and cancer type-specific peak sets from recent large-scale studies where TF binding sites were assessed in hematopoietic lineages and tumor samples from The Cancer Genome Atlas (TCGA) using ATAC-seq (Corces et al. [2016] Nat Genet 48:1193-1203; Corces et al. [2018] Science 362:eaav1898). As a result of these efforts, we generated comprehensive lists of high-confidence TFs and their respective accessibilities in hematopoietic lineages and three different epithelial lineages (i.e. breast, prostate, colon). This approach allowed us then to compare the accessibilities of the TFBSs as defined by ATAC-seq with those observed by us in the plasma DNA of patients with the respective epithelial malignancies. As outlined in the new version of the manuscript, we found excellent agreements.
- **More stringent statistical evaluation:** Furthermore, we realized that the presentation of our results in the first draft of the manuscript was not optimal and this caused reviewer #1 (the other two reviewers also expressed some concern about the figure quality) to describe the observed effects as weak and unconvincing. Therefore, we revised the entire statistical evaluation and description and accordingly also the presentation of the data. In the new version of the manuscript, we have considerably extended the explanation of the applied z-score statistics. In the literature, z-score statistic thresholds are usually set to ± 3 . To be more stringent and to demonstrate that we in fact do not observe weak effects, we raised the thresholds to ± 5 . We hope that with the improved explanation of our statistical approach and together with the newly designed figures that it now becomes obvious that the observed changes of TFBS accessibilities in plasma DNA are in fact strong effects.

- **Assessment of resolution limits:** As the issues of clinical utility and detection limit of the assay were raised and as detection of early-stage cancer represents a very current topic in the liquid biopsy community, we added substantial data to address this important issue accurately. To be more specific, we added the analyses of an additional cohort consisting of 592 plasma samples from individuals with colon cancer, where 82.1% of patients had stage I ($n=197$) or stage II ($n=280$) disease and compared those to 177 plasma samples from subjects with no current cancer diagnosis (for this newly added cohort the mean read count per sample was ~ 299.2 million, standard deviation 100.4 million reads). We compared accessibilities of colon/epithelial-specific TFBS GRHL2, EVX2, DLX2, HNF4A, LYL1, and PU.1 and observed already statistically significant different accessibilities between healthy controls and COAD samples at a 1% tumor level.
- **TF-based early cancer detection:** We used logistic regression with all 504 TFs to classify healthy controls from cancer samples and found that this approach is capable of identifying ctDNA in patients with stage I colon cancer with a 74% precision, 71% sensitivity and 72% specificity and patients with stage II colon cancer with a 84% precision, 74% sensitivity and 77% specificity. (The results are displayed in the new Figure 4). In the Discussion, we then compare our approach with other recently published liquid biopsy based methods aiming at early cancer detection.
- **Extended comparison to other nucleosomes positioning studies:** We extended the discussion section and clearly highlighted differences/improvements to the Snyder et al paper and our prior work (Ulz et al).
- **Newly designed figures.** We included completely newly designed figures with improved display. The new figures include among other things accessibility for selected TFBS from ATAC-seq data from TCGA, z-score distributions of cancer and control samples, z-scores of pooled plasma-Seq data for selected lineage specific TFBS, accessibilities of TFBS for neuroendocrine differentiation of additional prostate cancer samples, TFBS accessibilities in correlation to different tumor stages and tumor fractions, and the use of TFBS accessibilities for early cancer detection.

REVIEWERS' COMMENTS:

Reviewer #2 (Remarks to the Author):

Title: Inference of transcription factor binding from cell-free DNA enables tumor subtype prediction and detection of early-stage cancer

Authors: Ulz P et al

Summary: This is a revised manuscript that focuses on the analysis of cell-free DNA from cancer patients as an analyte for assessing the activity of transcription factors operative in tumors of various origins. The findings are of potentially very high impact, and provides a dimension beyond DNA mutation/copy number measurements that could provide insights into tumor diagnosis and therapy resistance using minimally invasive approaches.

1. The authors have provided detailed responses to each of the issues raised in the initial review of the manuscript/study. Further, additional data is included to support the conclusions and further extend confidence in the results.
2. Notable is the use of ATAC-seq to refine the list of high-confidence TFs by lineage.
3. The use of z-scores may still be problematic, but their use is explained in more detail with more stringent thresholds and should allow replication of the results.
4. The inclusion of early stage I/II colon cancer cohort is also quite interesting and highlights the potential for impact.
5. Minor point: The logistic regression analysis to classify early stage tumor is a nice demonstration of the approach, though the model is likely specific to this data and potentially less applicable as a for others to use on other data.

Reviewer #4 (Remarks to the Author): Recruited to replace Reviewer #1

The authors report analysis of WGS data for a massive number of cell-free DNA samples from cancer patients and healthy controls. They demonstrate that they can infer tumor and tumor subtype based on prediction of TF binding from nucleosome footprinting based on this data. A key outcome of this is the demonstration that they can detect early stage colorectal carcinomas with this approach.

This is a well written manuscript with the results clearly laid out.

Previous work from Jay Shendure's lab (Synder et al., Cell 2016) demonstrated that cfDNA sequencing can be used to infer cell types based on nucleosome footprinting. The extension of that work here is mainly in scope. Nevertheless, the demonstration that cfDNA sequencing can be used to predict tumor subtype and early stage cancer progression is an advance and is likely to be of clinical relevance.

In the discussion section highlighting the difference of this work from the Synder paper, the authors write that "Synder and colleagues did not profile individual TFs ..." However, the Synder paper does show TF footprinting for CTCF and a couple of other ubiquitous TFs. The extension of this manuscript is more in the prediction of cell-type-specific binding sites.

Lastly, as a mostly semantic issue, the authors define an "accessibility score" based on the amplitude of the high-frequency (non promoter) signals from the nucleosome occupancy patterns. This score is really more of a measure of the strength of nucleosome phasing at particular binding sites and related to the 'strength' of TF binding (or the fraction of cells with binding). Some discussion of this in the manuscript would aid in the interpretation of the data.

Reviewer #5 (Remarks to the Author): Recruited to replace Reviewer #3

In the revised version of the manuscript, Ulz et al have addressed the points risen satisfactorily.

In addition to providing the number of reads obtained per sample, it would be useful, however, to also provide the mean coverage in the Results section.

Reviewer #2

Summary: This is a revised manuscript that focuses on the analysis of cell-free DNA from cancer patients as an analyte for assessing the activity of transcription factors operative in tumors of various origins. The findings are of potentially very high impact, and provides a dimension beyond DNA mutation/copy number measurements that could provide insights into tumor diagnosis and therapy resistance using minimally invasive approaches.

1. The authors have provided detailed responses to each of the issues raised in the initial review of the manuscript/study. Further, additional data is included to support the conclusions and further extend confidence in the results.

2. Notable is the use of ATAC-seq to refine the list of high-confidence TFs by lineage.

3. The use of z-scores may still be problematic, but their use is explained in more detail with more stringent thresholds and should allow replication of the results.

4. The inclusion of early stage I/II colon cancer cohort is also quite interesting and highlights the potential for impact.

5. Minor point: The logistic regression analysis to classify early stage tumor is a nice demonstration of the approach, though the model is likely specific to this data and potentially less applicable as a for others to use on other data.

Our response: We appreciate this assessment by this referee. The point with the logistic regression analysis is well taken, but we believe that the strategy as outlined in the manuscript can be transferred to other data sets as well.

Reviewer #4: Recruited to replace Reviewer #1

The authors report analysis of WGS data for a massive number of cell-free DNA samples from cancer patients and healthy controls. They demonstrate that they can infer tumor and tumor subtype based on prediction of TF binding from nucleosome footprinting based on this data. A key outcome of this is the demonstration that they can detect early stage colorectal carcinomas with this approach.

This is a well written manuscript with the results clearly laid out.

Our response: We are grateful for these encouraging remarks.

Previous work from Jay Shendure's lab (Snyder et al., Cell 2016) demonstrated that cfDNA sequencing can be used to infer cell types based on nucleosome footprinting. The extension of that work here is mainly in scope. Nevertheless, the demonstration that cfDNA sequencing can be used to predict tumor subtype and early stage cancer progression is an advance and is likely to be of clinical relevance.

In the discussion section highlighting the difference of this work from the Snyder paper, the authors write that "Snyder and colleagues did not profile individual TFs ..." However, the Snyder paper does show TF footprinting for CTCF and a couple of other ubiquitous TFs. The extension of this manuscript is more in the prediction of cell-type-specific binding sites.

Our response: This point is well taken. We clarified this point in the discussion and made the changed the respective sentence on page 9 to: "First, Snyder and colleagues profiled only a small number of ubiquitous TFs ($n=8$) whereas we utilized recently generated ATAC-seq data and profiled numerous individual TFs, which enabled us to establish lineage-specific TFs for clinical applications."

Lastly, as a mostly semantic issue, the authors define an "accessibility score" based on the amplitude of the high-frequency (non promoter) signals from the nucleosome occupancy patterns. This score is really more of a measure of the strength of nucleosome phasing at particular binding sites and related to the 'strength' of TF binding (or the fraction of cells with binding). Some discussion of this in the manuscript would aid in the interpretation of the data.

Our response: We highly appreciate this comment. We completely agree with this referee and think that this thought is a very valuable addition to the discussion. Accordingly, we added this point, almost literally to the discussion on page 9.

Reviewer #5: Recruited to replace Reviewer #3

In the revised version of the manuscript, Ulz et al have addressed the points risen satisfactorily.

Our response: We are very happy with this assessment.

In addition to providing the number of reads obtained per sample, it would be useful, however, to also provide the mean coverage in the Results section.

Our response: We agree with this referee that the mean coverage is very important. We had included these values in Supplementary Table 4; however, we realized that this table was inadequately referenced in the main text. This Table is now referenced on page 4, where we describe the normal samples and the cases with prostate cancer (P40, P147, P148, P190), the colorectal adenocarcinoma (C2), and the two breast cancers (B7 and B13); and in addition on page 6, where we discuss the analyses of the pooled plasma samples. Regarding the 592 plasma samples from individuals with colon cancer we realized that we mentioned on page 8 only the mean read count per sample and we added the mean coverage at this location as well.